# A meta-analysis of genome-wide association studies identifies multiple longevity genes

Joris Deelen (ID) et al.#

Human longevity is heritable, but genome-wide association (GWA) studies have had limited success. Here, we perform two meta-analyses of GWA studies of a rigorous longevity phenotype definition including 11,262/3484 cases surviving at or beyond the age corresponding to the 90th/99th survival percentile, respectively, and 25,483 controls whose age at death or at last contact was at or below the age corresponding to the 60th survival percentile. Consistent with previous reports, rs429358 (apolipoprotein E (ApoE) ε4) is associated with lower odds of surviving to the 90th and 99th percentile age, while rs7412 (ApoE ε2) shows the opposite. Moreover, rs7676745, located near *GPR78*, associates with lower odds of surviving to the 90th percentile age. Gene-level association analysis reveals a role for tissue-specific expression of multiple genes in longevity. Finally, genetic correlation of the longevity GWA results with that of several disease-related phenotypes points to a shared genetic architecture between health and longevity.

Correspondence and requests for materials should be addressed to J.D. (email: Joris.Deelen@age.mpg.de) or to D.S.E. (email: DEvans@sfcc-cpmc.net) or to P.E.S. (email: P.Slagboom@lumc.nl) or to J.M.M. (email: murabito@bu.edu). #A full list of authors and their affiliations appears at the end of the paper.

The average human life expectancy has been increasing for centuries[1]. Based on twin studies, the heritability of human lifespan has been estimated to be ~25%, although this estimate differs among studies[2]. On the other hand, the heritability of lifespan based on the correlation of the mid-parent (i.e., the average of the father and mother) and offspring difference between age at death and expected lifespan was estimated to be 12%[3]. A recent study has indicated that the different heritability estimates may be inflated due to assortative mating, leaving a true heritability that is below 10%[4]. The heritability of lifespan, estimated using the sibling relative risk, increases with age[5] and is assumed to be enriched in long-lived families, particularly when belonging to the 10% longest-lived of their generation[6]. To identify genetic associations with human lifespan, several genome-wide association (GWA) studies have been performed[7–20]. These studies have used a discrete (i.e., older cases versus younger controls) or a continuous phenotype (such as age at death of individuals or their parents). The selection of cases for the studies using a discrete longevity phenotype has been based on the survival to ages above 90 or 100 years or belonging to the top 10% or 1% of survivors in a population. Studies defining cases using a discrete longevity phenotype often need to rely on controls from more contemporary birth cohorts, because all others from the case birth cohorts have died before sample collection. Previous GWA studies have identified several genetic variants, but the only locus that has shown genome-wide significance ($P \leq 5 \times 10^{-8}$) in multiple independent meta-analyses of GWA studies is apolipoprotein E (*APOE*)[21], where the ApoE ε4 variant is associated with lower odds of being a long-lived case.

The lack of replication for many reported associations with longevity could be due, at least partly, to the use of different definitions for cases and controls between studies. Furthermore, even within a study, the use of a single age cut-off phenotype for men and women and for individuals belonging to different birth cohorts will give rise to heterogeneity, as survival probabilities differ by sex and birth cohort[22], and genetic effects are known to be age- and birth cohort-specific[5,23]. In an attempt to mitigate the effects of heterogeneous case and control groups, we use country-, sex- and birth cohort-specific life tables to identify ages that correspond to different survival percentiles to define cases and controls in our meta-analyses of GWA studies of longevity. Furthermore, most studies in our meta-analyses use controls from the same study population as the cases, which limits the impact of sampling biases that could confound associations. The current meta-analyses include individuals from 20 cohorts from populations of European, East Asian, or African American descent. Two sets of cases are examined: individuals surviving at or beyond the age corresponding to the 90th survival percentile (90th percentile cases) or the 99th survival percentile (99th percentile cases) based on life tables specific to the country where each cohort was based, sex, and birth cohort (i.e., birth year). The same country-, sex-, and birth cohort-specific life tables are used to define the age threshold for controls, corresponding to the 60th percentile of survival. We identify two genome-wide significant loci, of which one is replicated in two independent European cohorts that use de novo genotyping. We also perform a gene-level association analysis based on tissue-specific gene expression and identify additional longevity genes. In addition, using linkage disequilibrium (LD) score regression[24], we show that longevity is genetically correlated with multiple diseases and traits.

## Results

**Genome-wide association meta-analyses.** We performed two meta-analyses in individuals of European ancestry combining cohort-specific genome-wide association data generated using 1000 Genomes imputation: (1) 90th percentile cases versus all controls and (2) 99th percentile cases versus all controls. The numbers of cases and controls in each study are shown in Table 1. For both case definitions, multiple genetic variants at the well-replicated *APOE* locus reached genome-wide significance ($P \leq 5 \times 10^{-8}$) (Table 2, Fig. 1 and Supplementary Fig. 1). Consistent with previous reports, rs429358 (ApoE ε4) was associated with lower odds of surviving to the 90th or 99th percentile age at the genome-wide significance level. In addition, we report a genome-wide significant association of rs7412 (ApoE ε2) with higher odds of surviving to the 90th and the 99th percentile age. Conditional analysis in two of the cohorts with individuals of European ancestry, CEPH and LLS (combined with GEHA Dutch) (representing 18% of the 90th percentile cases and 6% of all controls), indicated that the signal at the *APOE* locus was explained by these two independent variants, i.e., rs429358 (ApoE ε4) and rs7412 (ApoE ε2). There was no evidence of heterogeneity of effect across cohorts for ApoE ε2 ($P$-value for heterogeneity ($P_{\text{het}}$) = 0.619, Table 2). For ApoE ε4, on the other hand, there was evidence of heterogeneity ($P_{\text{het}} = 0.004$, Table 2), although the direction of effect of this variant was consistent across cohorts (Fig. 2). Besides ApoE ε4 and ε2, one additional variant, rs7676745, located on chromosome 4 near *GPR78*, showed a genome-wide significant association in the 90th percentile cases versus all controls analysis ($P = 4.3 \times 10^{-8}$, Table 2). The rare allele of this variant (A) was associated with lower odds of surviving to the 90th percentile age and there was no evidence of heterogeneity of effect across cohorts ($P_{\text{het}} = 0.462$, Table 2). The regional association and forest plots for this locus are depicted in Figs. 1 and 2.

Most of the variants reported in Table 2 show stronger effects in the 99th percentile as compared to the 90th percentile analysis (Supplementary Fig. 2), indicating that the use of a more extreme phenotype results in stronger effects.

**Replication.** The effects of ApoE ε4 and ε2 were replicated in the two cohorts (i.e., DKLSII and GLS) in which de novo genotyping, using predesigned Taqman SNP Genotyping Assays, was applied (Table 2). However, we were not able to replicate the effect of rs7676745 in these cohorts, since there was no Taqman SNP Genotyping Assay available for this variant.

**Validation in parental age-based data sets.** Given that all available studies with genome-wide genetic data that met our inclusion criteria were included in our genome-wide association meta-analyses, we additionally set out to validate our findings in two UK Biobank parental longevity data sets (Table 1) and the parental lifespan data set recently created by Timmers and colleagues[20]. Since the genotyped individuals in the UK Biobank were recruited at relatively young ages (40–69 years), these data sets were based on the age reached by the parents of the study participants. Hence, the phenotypes used for validation were different from those used in our meta-analyses, resulting in smaller effect sizes. Moreover, the reference panels used to impute the genetic variants (a merged panel of UK10K, 1000G Phase 3, and Haplotype Reference Consortium (HRC) for parental longevity and HRC alone for parental lifespan)[20] were different from the one used in our meta-analyses (1000G Phase 1), which could have influenced the outcome of the analyses. Of the variants that showed a $P$-value $\leq 1 \times 10^{-6}$ in our meta-analyses (Table 2), only ApoE ε4 and ε2 were significantly associated with both parental longevity and lifespan ($P < 0.05$) in these data sets (Table 3). Moreover, the rare allele (A) of the second most significant

**Table 1 Samples included in the different genome-wide association meta-analyses or the replication and validation**

| Study | Ancestry | 90th percentile cases | 99th percentile cases | All controls | Dead controls |
|---|---|---|---|---|---|
| Discovery | | | | | |
| 100-plus/LASA/ADC | European | 373 | 301 | 2271 | 245 |
| AGES | European | 300 | | 1001 | 466 |
| CEPH[a] | European | 1234 | 1112 | 831 | |
| CHS | European | 905 | 68 | 558 | 539 |
| DKLS[a] | European | 960 | 610 | 1917 | |
| FHS | European | 332 | | 1444 | 539 |
| GEHA Danish[a] | European | 451 | 127 | 900 | |
| GEHA French | European | 271 | 81 | 358 | |
| GEHA Italy | European | 182 | | 184 | |
| HRS | European | 361 | | 3312 | 657 |
| LLFS | European | 1110 | 339 | 552 | 82 |
| LLS + GEHA Dutch | European | 1037 | 377 | 712 | |
| Longevity | European | 548 | 271 | 584 | |
| MrOS | European | 1171 | 82 | 386 | 320 |
| Newcastle 85 + [a] | European | 215 | | 5159 | |
| RS | European | 774 | 79 | 2965 | 1731 |
| SOF | European | 812 | 37 | 354 | 300 |
| Vitality 90 + [a] | European | 226 | | 1995 | |
| Total | | 11,262 | 3484 | 25,483 | 4879 |
| Replication | | | | | |
| DKLSII[a] | European | 944 | 298 | 772 | |
| GLS | European | 1613 | 1613 | 4215 | |
| Total | | 2557 | 1911 | 4987 | |
| Validation | | | | | |
| UK Biobank | European | 19,742 | 928 | 19,698 | |
| Trans-ethnic | | | | | |
| CLHLS | East Asian | 2178 | 2178 | 2299 | |
| CHS | African American | 177 | | 211 | |
| Total | | 13,617 | 5662 | 27,993 | |

*100-plus* 100-plus Study, *LASA* Longitudinal aging study of Amsterdam, *ADC* Amsterdam dementia cohort, *AGES* Age/Gene Environment Susceptibility Study, *CEPH* CEPH centenarian cohort, *CHS* Cardiovascular Health Study, *FHS* Framingham Heart Study, *GEHA* Genetics of Healthy Aging Study, *HRS* Health and Retirement Study, *LLFS* Long Life Family Study, *LLS* Leiden Longevity Study, *Longevity* Longevity Gene Project, *MrOS* Osteoporotic Fractures in Men Study, *Newcastle 85 +* Newcastle 85 + Study, *RS* Rotterdam study, *SOF* Study of Osteoporotic Fracture, *Vitality 90 +* Vitality 90 + project, *GLS* German longevity study, *CLHLS* Chinese Longitudinal Healthy Longevity Survey
[a]For these studies, controls were provided by a separate cohort. Further details of the cohorts are provided in Supplementary Data 4

**Table 2 Results of the European genome-wide association meta-analyses and replication in the de novo genotyped cohorts**

| rsID | Chr:Position | Candidate/ closest gene | Alleles (EA/ OA) | EAF | OR | 95% CI | P | I2 (%) | Phet |
|---|---|---|---|---|---|---|---|---|---|
| 90th percentile cases versus all controls (Discovery) | | | | | | | | | |
| rs116362179 | 2:53,380,757 | — | T/C | 0.05 | 1.34 | 1.20–1.50 | $4.9 \times 10^{-7}$ | 0 | 0.457 |
| rs7676745[a] | 4:8,565,547 | GPR78 | A/G | 0.04 | 0.67 | 0.57–0.77 | $4.3 \times 10^{-8}$ | 0 | 0.462 |
| rs7754015 | 6:127,206,068 | — | G/T | 0.43 | 0.90 | 0.86–0.94 | $6.8 \times 10^{-7}$ | 0 | 0.670 |
| rs35262860 | 8:55,478,909 | RP1 | GCT/G | 0.39 | 1.11 | 1.07–1.15 | $3.9 \times 10^{-7}$ | 0 | 0.941 |
| rs3138136 | 12:56,117,570 | RDH5 | T/C | 0.10 | 0.83 | 0.77–0.89 | $5.4 \times 10^{-7}$ | 14.5 | 0.284 |
| rs429358 | 19:45,411,941 | APOE | C/T | 0.13 | 0.60 | 0.56–0.64 | $1.3 \times 10^{-56}$ | 54.3 | 0.004 |
| rs7412 | 19:45,412,079 | APOE | T/C | 0.09 | 1.28 | 1.19–1.37 | $2.4 \times 10^{-11}$ | 0 | 0.619 |
| 90th percentile cases versus all controls (Replication) | | | | | | | | | |
| rs429358 | 19:45,411,941 | APOE | C/T | | 0.45 | 0.40–0.51 | $5.2 \times 10^{-36}$ | 85.4 | 0.009 |
| rs7412 | 19:45,412,079 | APOE | T/C | | 1.32 | 1.18–1.48 | $2.4 \times 10^{-6}$ | 16.6 | 0.274 |
| 99th percentile cases versus all controls (Discovery) | | | | | | | | | |
| rs3830412 | 3:124,397,321 | KALRN | A/AT | 0.22 | 1.21 | 1.12–1.30 | $4.3 \times 10^{-7}$ | 0 | 0.767 |
| rs138762279 | 5:173,710,197 | — | AT/A | 0.16 | 0.79 | 0.72–0.86 | $1.2 \times 10^{-7}$ | 0 | 0.769 |
| rs62502826 | 8:28,982,295 | KIF13B | A/G | 0.15 | 1.23 | 1.13–1.33 | $5.6 \times 10^{-7}$ | 14.9 | 0.298 |
| rs7039467 | 9:22,056,213 | CDKN2A/B | A/G | 0.48 | 1.20 | 1.12–1.28 | $1.1 \times 10^{-7}$ | 0 | 0.843 |
| rs429358 | 19:45,411,941 | APOE | C/T | 0.13 | 0.52 | 0.47–0.58 | $3.9 \times 10^{-34}$ | 0 | 0.833 |
| rs7412 | 19:45,412,079 | APOE | T/C | 0.09 | 1.47 | 1.32–1.64 | $3.2 \times 10^{-12}$ | 0 | 0.639 |
| 99th percentile cases versus all controls (Replication) | | | | | | | | | |
| rs429358 | 19:45,411,941 | APOE | C/T | | 0.44 | 0.38–0.50 | $4.0 \times 10^{-32}$ | 84.0 | 0.012 |
| rs7412 | 19:45,412,079 | APOE | T/C | | 1.35 | 1.19–1.53 | $2.0 \times 10^{-6}$ | 0 | 0.534 |

*EA* effect allele, *OA* other allele, *EAF* effect allele frequency, *OR* odds ratio (i.e., odds to become long-lived when carrying the effect allele); *95% CI* 95% confidence interval, *I2* heterogeneity statistic, $P_{het}$ P-value for heterogeneity
[a]We were not able to replicate the effect of this genetic variant, since there was no Taqman SNP Genotyping Assay available. We only report the most significant genetic variant for the loci with at least one variant with a P-value ≤ 1 × 10⁻⁶. The rsID is based on dbSNP build 150. The *Chr:Position* is based on Genome Reference Consortium Human Build 37 (GRCh37)

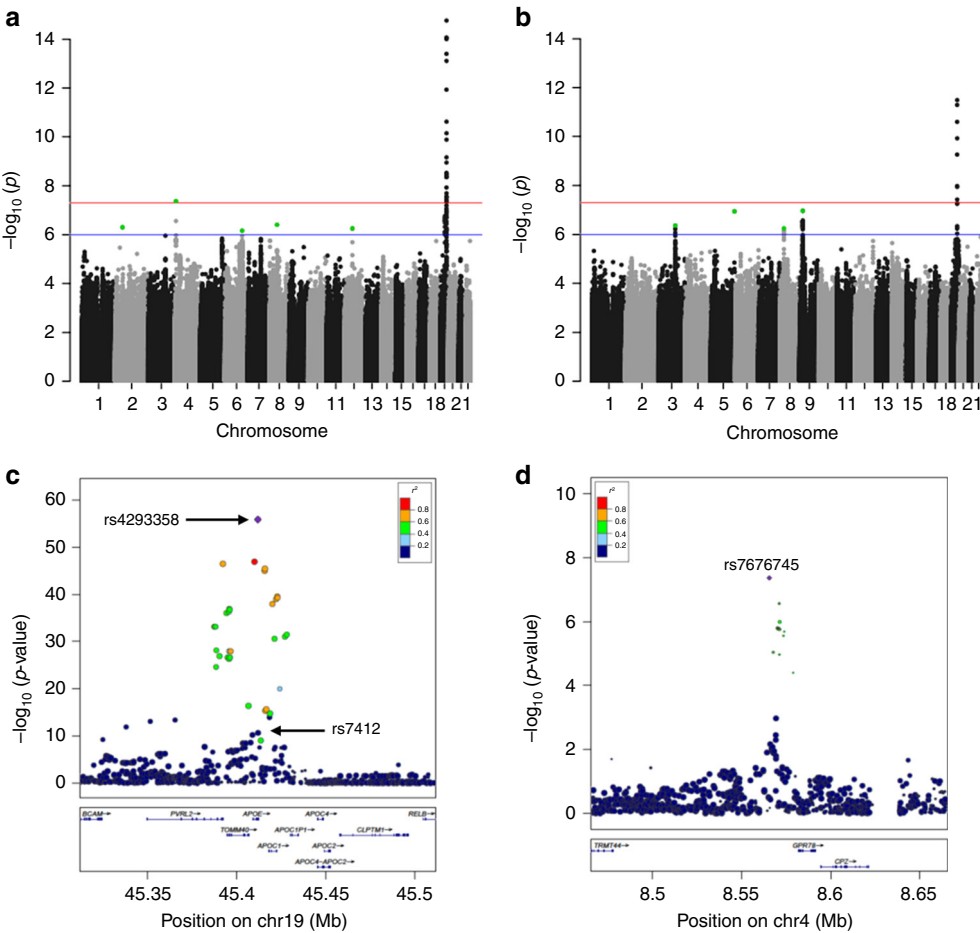

**Fig. 1** Results of the European genome-wide association meta-analyses. Manhattan plot presenting the –log$_{10}$ $P$-values from the European genome-wide association meta-analysis of the 90th percentile cases versus all controls (**a**) and 99th percentile cases versus all controls (**b**). The red line indicates the threshold for genome-wide significance ($P \leq 5 \times 10^{-8}$), while the blue line indicates the threshold for genetic variants that showed a suggestive significant association ($P \leq 1 \times 10^{-6}$). The variants that are reported in Table 2 are highlighted in green. For representation purposes, the maximum of the y-axis was set to 14. Regional association plot for the *APOE* (**c**) and *GPR78* (**d**) loci based on the results from the 90th percentile cases versus all controls meta-analysis. The colour of the variants is based on the linkage disequilibrium with rs429358 (ApoE ε4) (**c**) or rs7676745 (**d**)

variant at the *CDKN2A/B* locus, rs2184061, was associated with increased parental lifespan ($P = 8.4 \times 10^{-6}$), but not with parental longevity ($P = 0.329$). However, we had adequate power to validate all of our identified variants, even when the effect sizes were halved in the parental longevity data sets.

**Trans-ethnic meta-analyses.** We subsequently performed two trans-ethnic meta-analyses (90th and 99th percentile cases versus all controls) to see if the increase in sample size would lead to identification of additional longevity loci. In this analysis we included individuals of European (all previously used data sets), East Asian (CLHLS), and African American (CHS) ancestry. However, with the exception of *APOE* and rs2069837, located in *IL6*, which has previously been associated with longevity in CLHLS[9], this analysis did not identify additional genome-wide significant loci (Table 4, Fig. 3 and Supplementary Fig. 3). The observed association of the genetic variant in *IL6* in the trans-ethnic meta-analyses was mainly driven by the association in the East Asian population. The other variant previously associated with longevity in CLHLS[9], rs2440012, located in *ANKRD20A9P*, did not pass quality control in the large majority of the included cohorts from populations of European descent and was thus not analysed in the trans-ethnic meta-analyses.

**Comparison of control definitions.** To examine the impact of the definition of controls, we performed a sensitivity analysis in which we compared the results of the meta-analysis using the same case definition (90th percentile) with (1) all controls and (2) dead controls only. For this analysis, only cohorts that contributed results using both control definitions were considered (i.e., 100-plus/LASA/ADC, AGES, CHS, FHS, HRS, LLFS, MrOS, RS, and SOF). The results of the two meta-analyses with different control groups were very similar (Supplementary Fig. 4). Among the three loci with at least one genetic variant with a $P$-value $\leq 1 \times 10^{-6}$ in either meta-analysis (and analysed in the same cohorts in both meta-analyses), the most significant variants had odds ratios (ORs) that differed by <1% (Supplementary Table 1).

**Replication of previously identified loci for human lifespan.** To determine the association of previously identified loci for human lifespan and longevity, we performed a look-up of the reported genetic variants within these loci in our meta-analyses data sets. The only previously identified loci that contained variants that showed a significant ($P < 7.8 \times 10^{-4}$, i.e., Bonferroni adjusted for the number of tested loci ($n = 64$)) and directionally consistent associations in our study were *FOXO3* and *CDKN2A/B* (Supplementary Data 1). As depicted in Supplementary Fig. 5, the

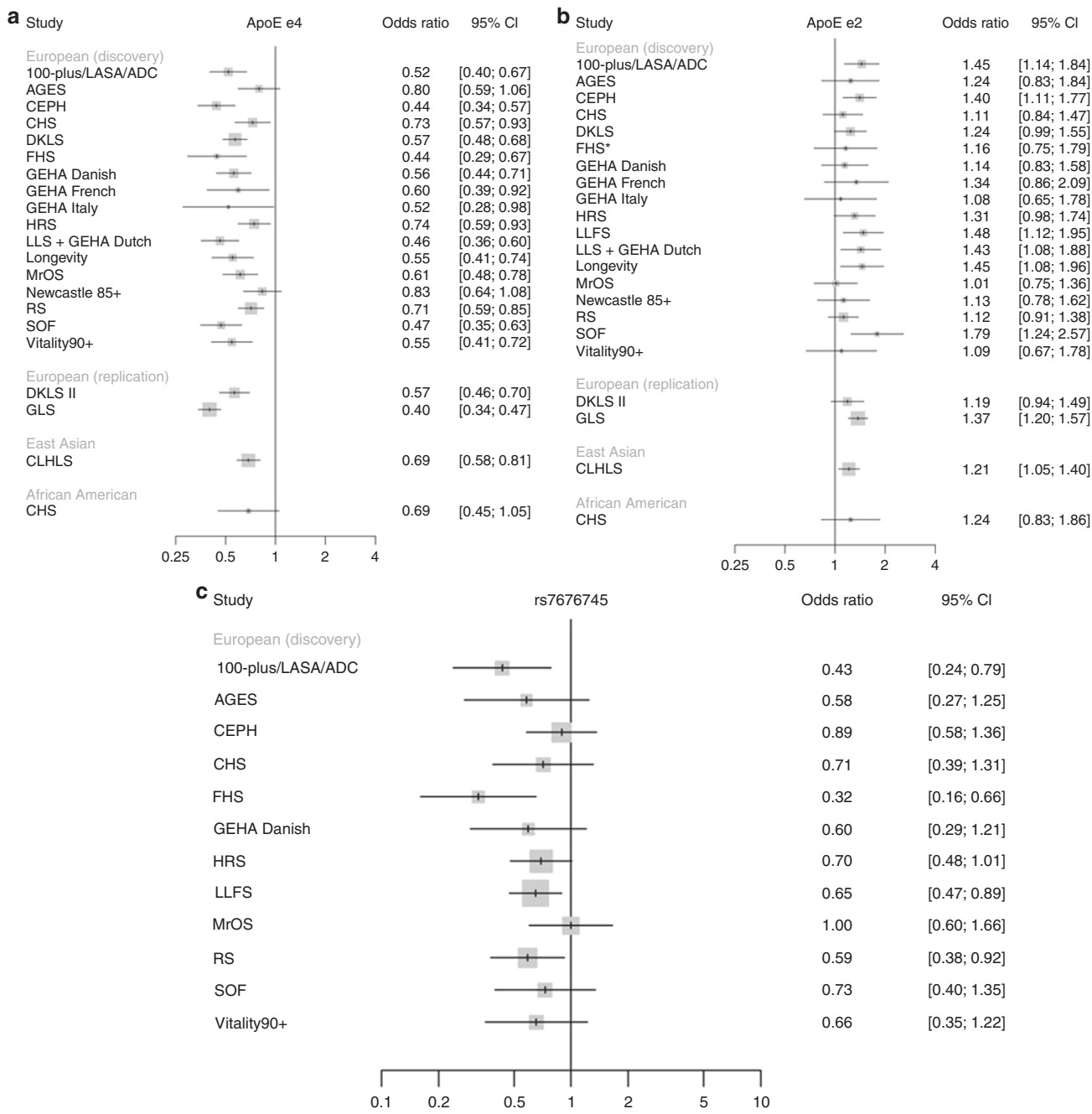

**Fig. 2** Study-specific results for the genetic variants in *APOE* and *GPR78*. Forest plots for the ApoE ε4 (**a**) and ε2 (**b**) variants and rs7676745 (**c**) based on the results from the 90th percentile versus all controls analysis. The size of the boxes represents the sample size of the cohort. We had no data available for ApoE ε4 in LLFS and for rs7676745 in DKLS, GEHA Italy, GEHA Danish, LLS (combined with GEHA Dutch), Longevity, and Newcastle 85 + . The data for ApoE ε2 in FHS was based on imputation using the Haplotype Reference Consortium reference panel due to the low-imputation quality of this variant when using the 1000 Genomes reference panel

effects of the most frequently reported variants within these loci (i.e., rs2802292 and rs1556516) fluctuate between cohorts and there seems to be no correlation with the genetic background of the included populations. However, for the reported variants within both loci, the odds of surviving to the 99th percentile age is higher than the odds of surviving to the 90th percentile age, indicating they likely affect both early and late-life mortality.

Several of the loci that have been associated with increased parental lifespan in the most recent and largest meta-analysis of GWA studies for this phenotype (i.e., *KCNK3*, *HTT*, *LPA*, *ATXN2/BRAP*, and *LDLR*)[20] contain genetic variants that show a nominal significant association ($P < 0.05$) with higher odds of

surviving to the 90th and/or 99th percentile age. Since the phenotypes used in our study (i.e., cases surviving at or beyond the age corresponding to the 90th/99th survival percentile) were different from the one used in the previous study (i.e., parental lifespan), we performed an additional look-up of these variants in one of the UK Biobank data sets we created for validation of our findings (i.e., the 90th percentile cases versus all controls data set). With the exception of the variant in *HTT*, all variants showed a nominal significant association in this data set (Supplementary Table 2), indicating that the lack of significant replication of these loci in our discovery phase data set is not likely to be due to a difference in the used phenotype.

**Table 3 Results of the validation in the UK Biobank parental age-based data sets**

| rsID | Chr:Position | Candidate/ closest gene | Alleles (EA/OA) | EAF | OR | 95% CI | *P* |
|------|-------------|------------------------|-----------------|-----|-----|--------|-----|
| 90th percentile cases versus all controls (Parental longevity) | | | | | | | |
| rs116362179 | 2:53,380,757 | — | T/C | 0.04 | 1.01 | 0.94–1.08 | 0.775 |
| rs7676745 | 4:8,565,547 | GPR78 | A/G | 0.04 | 0.98 | 0.92–1.06 | 0.667 |
| rs7754015 | 6:127,206,068 | — | G/T | 0.43 | 1.00 | 0.97–1.03 | 0.832 |
| rs35262860 | 8:55,478,909 | RP1 | GCT/G | 0.39 | 0.97 | 0.94–0.99 | 0.021 |
| rs3138136 | 12:56,117,570 | RDH5 | T/C | 0.11 | 1.00 | 0.95–1.04 | 0.863 |
| rs429358 | 19:45,411,941 | APOE | C/T | 0.16 | 0.85 | 0.81–0.88 | $1.1 \times 10^{-16}$ |
| rs7412 | 19:45,412,079 | APOE | T/C | 0.08 | 1.12 | 1.06–1.18 | $2.2 \times 10^{-5}$ |
| 90th percentile cases versus all controls (Parental lifespan) | | | | | | | |
| rs116362179 | 2:53,380,757 | — | T/C | 0.04 | 1.00 | 0.98–1.02 | 0.697 |
| rs7676745 | 4:8,565,547 | GPR78 | A/G | 0.05 | 1.01 | 0.99–1.03 | 0.247 |
| rs3138136 | 12:56,117,570 | RDH5 | T/C | 0.11 | 0.99 | 0.98–1.00 | 0.135 |
| rs429358 | 19:45,411,941 | APOE | C/T | 0.15 | 0.90 | 0.89–0.91 | $3.1 \times 10^{-83}$ |
| rs7412 | 19:45,412,079 | APOE | T/C | 0.08 | 1.06 | 1.05–1.08 | $7.6 \times 10^{-17}$ |
| 99th percentile cases versus all controls (Parental longevity) | | | | | | | |
| rs3830412 | 3:124,397,321 | KALRN | A/AT | 0.20 | 1.11 | 0.99–1.24 | 0.081 |
| rs138762279 | 5:173,710,197 | — | AT/A | 0.34 | 1.05 | 0.95–1.17 | 0.299 |
| rs62502826 | 8:28,982,295 | KIF13B | A/G | 0.14 | 1.04 | 0.90–1.19 | 0.614 |
| rs7039467 | 9:22,056,213 | CDKN2A/B | A/G | 0.69 | 0.93 | 0.83–1.05 | 0.245 |
| rs2184061 | 9:22,061,562 | CDKN2A/B | A/C | 0.40 | 0.95 | 0.87–1.05 | 0.329 |
| rs429358 | 19:45,411,941 | APOE | C/T | 0.16 | 0.76 | 0.66–0.87 | $9.6 \times 10^{-5}$ |
| rs7412 | 19:45,412,079 | APOE | T/C | 0.08 | 1.23 | 1.05–1.45 | 0.011 |
| 99th percentile cases versus all controls (Parental lifespan) | | | | | | | |
| rs62502826 | 8:28,982,295 | KIF13B | A/G | 0.14 | 1.00 | 0.99–1.02 | 0.376 |
| rs2184061 | 9:22,061,562 | CDKN2A/B | A/C | 0.40 | 1.02 | 1.01–1.03 | $8.4 \times 10^{-6}$ |
| rs429358 | 19:45,411,941 | APOE | C/T | 0.15 | 0.90 | 0.89–0.91 | $3.1 \times 10^{-84}$ |
| rs7412 | 19:45,412,079 | APOE | T/C | 0.08 | 1.06 | 1.05–1.08 | $7.6 \times 10^{-17}$ |

For the *CDKN2A/B* locus we have also reported the second most significant variant in this locus (rs2184061), since the allele frequency of the most significant variant (rs7039467) is not comparable between the meta-analyses and UK Biobank data sets due to difference in the reference panel used for imputation. The *rsID* is based on dbSNP build 150. The *Chr:Position* is based on Genome Reference Consortium Human Build 37 (GRCh37)
*EA* effect allele, *OA* other allele, *EAF* effect allele frequency, *OR* odds ratio (i.e., odds of parent(s) to become long-lived when carrying the effect allele), *95% CI* 95% confidence interval

**Gene-level association analysis**. In addition to genetic variant associations, GWA studies can also be used to identify gene-level associations by integrating results from expression quantitative trait locus (eQTL) studies that relate variants to gene expression. In order to identify gene-level associations, we used MetaXcan, an analytic approach that uses tissue-specific eQTL results from the GTEx project to estimate gene-level associations with the trait examined from summary-level GWA study results[25]. Tissue-specific genetically predicted expression of 14 genes (*ANKRD31*, *BLOC1S1*, *KANSL1*, *CRHR1*, *ARL17A*, *LRRC37A2*, *ERCC1*, *RELB*, *DMPK*, *CD3EAP*, *PVRL2*, *GEMIN7*, *BLOC1S3*, and *APOC2*) was significantly associated with survival to the 90th and/or 99th percentile age after adjustment for multiple testing (Table 5). Eight of these genes (*ERCC1*, *RELB*, *DMPK*, *CD3EAP*, *PVRL2*, *GEMIN7*, *BLOC1S3*, and *APOC2*) are located near the *APOE* gene, raising the likely possibility that these associations reflected the influence of variants in this well-established longevity-associated locus. The remaining genes are located on chromosome 5, 12, and 17. As depicted in Supplementary Data 2, distinct sets of genetic variants were used by MetaXcan for all significant tissue-specific gene expression associations with survival to the 90th and/or 99th percentile age.

**Genetic correlation analyses**. LD score regression was performed to determine the genetic correlation between the different case definitions used for our meta-analyses (based on the results from the European cohorts only), and between longevity and other traits and diseases[24]. The genetic correlation (rg) between the 90th and 99th percentile analysis, using all controls for both groups, was 1.01 (SE = 0.06, $P = 3.9 \times 10^{-66}$). Using LD Hub[26], which performs automated LD score regression, we subsequently

estimated the genetic correlation of our phenotypes with 246 diseases and traits available in their database. We found a significant genetic correlation of our phenotypes with the father's age at death phenotype from the UK Biobank. The most significant (negative) genetic correlation of both our phenotypes was with coronary artery disease (CAD) (rg (SE) = −0.40 (0.07) and rg (SE) = −0.29 (0.07), respectively) and several traits involved in type 2 diabetes (T2D) also showed a significant association with one or both phenotypes after Bonferroni adjustment for multiple testing (Table 6 and Supplementary Data 3).

**Discussion**
We brought together studies from all over the world to perform GWA study meta-analyses in over 13,000 long-lived individuals of diverse ethnic background, including European, East Asian and African American ancestry, to characterise the genetic architecture of human longevity. We used the 1000 Genomes reference panel for imputation to expand the coverage of the genome in comparison to previous GWA studies of longevity. Consistent with previous reports, rs429358, defining ApoE ε4, was associated with decreased odds of becoming long-lived. Moreover, we report a genome-wide significant association of rs7412, defining ApoE ε2, with increased odds of becoming long-lived. We additionally found a genome-wide significant association of a locus near *GPR78*. Gene-level association analysis revealed association of increased *KANSL1*, *CRHR1*, *ARL17A*, and *LRRC37A2* expression and decreased *ANKRD31* and *BLOC1S1* expression with increased odds of becoming long-lived. Genetic correlation analysis showed that our longevity phenotypes are genetically correlated with father's age at death, CAD and T2D-related phenotypes.

**Table 4 Results of the trans-ethnic genome-wide association meta-analyses**

| rsID | Chr:Position | Candidate/ closest gene | Alleles (EA/OA) | EAF | OR | 95% CI | P | $I^2$ (%) | $P_{het}$ |
|---|---|---|---|---|---|---|---|---|---|
| 90th percentile cases versus all controls | | | | | | | | | |
| rs12143832 | 1:21,705,436 | ECE1 | C/T | 0.46 | 0.90 | 0.87–0.94 | $2.0 \times 10^{-7}$ | 0 | 0.722 |
| rs7676745 | 4:8,565,547 | GPR78 | A/G | 0.04 | 0.67 | 0.58–0.78 | $1.7 \times 10^{-7}$ | 1.8 | 0.428 |
| rs1262476 | 6:126,986,996 | — | A/G | 0.24 | 1.12 | 1.07–1.17 | $9.8 \times 10^{-7}$ | 0 | 0.574 |
| rs2069837 | 7:22,768,027 | IL6 | G/A | 0.08 | 0.90 | 0.82–0.99 | $5.2 \times 10^{-8}$ | 50.7 | 0.005 |
| rs35262860 | 8:55,478,909 | RP1 | GCT/G | 0.39 | 1.11 | 1.07–1.15 | $5.6 \times 10^{-7}$ | 0 | 0.955 |
| rs62127362 | 19:33,458,479 | CEP89 | C/G | 0.13 | 0.87 | 0.82–0.93 | $4.3 \times 10^{-7}$ | 21.4 | 0.190 |
| rs429358 | 19:45,411,941 | APOE | C/T | 0.13 | 0.60 | 0.55–0.66 | $1.0 \times 10^{-61}$ | 52.1 | 0.004 |
| rs7412 | 19:45,412,079 | APOE | T/C | 0.09 | 1.26 | 1.19–1.35 | $1.7 \times 10^{-12}$ | 0 | 0.718 |
| 99th percentile cases versus all controls | | | | | | | | | |
| rs2758603 | 1:156,198,994 | PMF1 | C/T | 0.34 | 1.12 | 1.02–1.22 | $9.8 \times 10^{-7}$ | 57.2 | 0.005 |
| rs3830412 | 3:124,397,321 | KALRN | A/AT | 0.22 | 1.21 | 1.12–1.30 | $8.2 \times 10^{-7}$ | 0 | 0.767 |
| rs138762279 | 5:173,710,197 | — | AT/A | 0.16 | 0.79 | 0.72–0.86 | $2.2 \times 10^{-7}$ | 0 | 0.769 |
| rs2069837 | 7:22,768,027 | IL6 | G/A | 0.09 | 0.90 | 0.76–1.08 | $1.4 \times 10^{-8}$ | 67.7 | $3.5 \times 10^{-4}$ |
| rs7039467 | 9:22,056,213 | CDKN2A/B | A/G | 0.48 | 1.20 | 1.12–1.28 | $2.1 \times 10^{-7}$ | 0 | 0.843 |
| rs429358 | 19:45,411,941 | APOE | C/T | 0.13 | 0.55 | 0.50–0.61 | $1.3 \times 10^{-36}$ | 20.0 | 0.247 |
| rs7412 | 19:45,412,079 | APOE | T/C | 0.09 | 1.39 | 1.26–1.53 | $1.7 \times 10^{-12}$ | 10.0 | 0.347 |

We only report the most significant genetic variant for the loci with at least one variant with a P-value ≤1 × 10⁻⁶. The reported P is the P-value from the Han-Eskin random-effects (RE2) model from METASOFT. The rsID is based on dbSNP build 150. The Chr:Position is based on Genome Reference Consortium Human Build 37 (GRCh37). EA effect allele, OA other allele, EAF effect allele frequency (based on individuals of European ancestry only), OR odds ratio (i.e., odds to become long-lived when carrying the effect allele), 95% CI 95% confidence interval, $P_{het}$ P-value for heterogeneity, $I^2$ heterogeneity statistic.

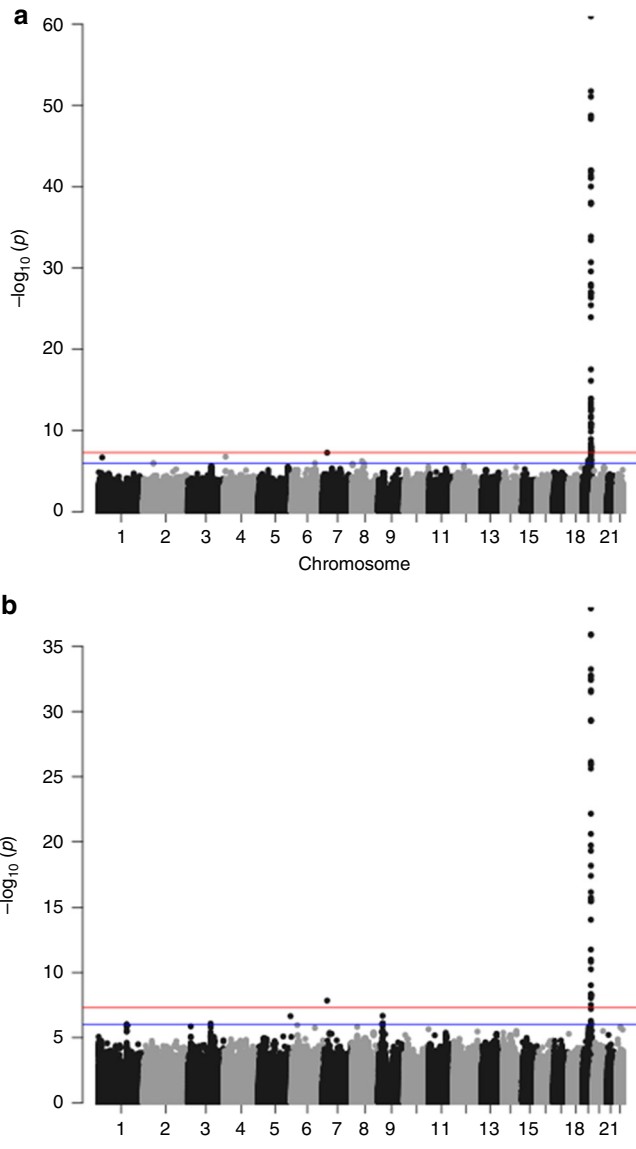

**Fig. 3** Results of the trans-ethnic genome-wide association meta-analyses. Manhattan plot presenting the –log₁₀ P-values from the trans-ethnic genome-wide association meta-analysis of the 90th percentile cases versus all controls (**a**) and 99th percentile cases versus all controls (**b**). The red line indicates the threshold for genome-wide significance ($P \leq 5 \times 10^{-8}$), while the blue line indicates the threshold for genetic variants that showed a suggestive significant association ($P \leq 1 \times 10^{-6}$)

Genetic variation in *APOE* is well known to be associated with longevity and lifespan, with the first report more than two decades ago in a small candidate gene study[27]. Since then, there have been numerous candidate gene studies, including individuals of diverse ancestry, which have identified associations of ApoE with longevity[28–32]. However, thus far, rs7412, the ApoE ε2-defining, genetic variant has not been reported to show a genome-wide significant association in GWA studies of longevity and lifespan. This could be due to the fact that we performed imputation using the 1000 Genomes reference panel, while earlier GWA studies used the HapMap reference panel, which has limited coverage of this variant. ApoE mediates cholesterol metabolism in peripheral tissues and is the principal cholesterol carrier in the brain. The ApoE ε2 and ε4 variants have previously been associated with a decreased (ε2) or increased (ε4) risk for several age-related

**Table 5 Results of the gene-level association analyses**

| Genes | Ensembl ID | Chromosome band | Tissue | $OR_{90}$ | $P_{90}$ | $OR_{99}$ | $P_{99}$ |
|---|---|---|---|---|---|---|---|
| ANKRD31 | ENSG00000145700 | 5q13.3 | Stomach | 0.63 | **$1.1 \times 10^{-6}$** | 0.61 | $9.0 \times 10^{-4}$ |
| BLOC1S1 | ENSG00000135441 | 12q13.2 | Adipose subcutaneous | 0.49 | **$4.5 \times 10^{-7}$** | 0.56 | 0.009 |
| KANSL1 | ENSG00000120071 | 17q21.31 | Skin sun exposed lower leg | 1.22 | **$1.5 \times 10^{-6}$** | 1.26 | $1.9 \times 10^{-4}$ |
| CRHR1 | ENSG00000120088 | 17q21.31 | Nerve tibial | 1.54 | **$3.4 \times 10^{-7}$** | 1.81 | $6.2 \times 10^{-6}$ |
| ARL17A | ENSG00000185829 | 17q21.31 | Artery aorta | 1.24 | **$8.1 \times 10^{-7}$** | 1.31 | $5.9 \times 10^{-5}$ |
| ARL17A | ENSG00000185829 | 17q21.31 | Breast mammary tissue | 1.18 | **$1.8 \times 10^{-6}$** | 1.22 | $3.2 \times 10^{-4}$ |
| ARL17A | ENSG00000185829 | 17q21.31 | Colon sigmoid | 1.21 | **$2.2 \times 10^{-6}$** | 1.21 | 0.002 |
| LRRC37A2 | ENSG00000238083 | 17q21.31 | Minor salivary gland | 1.17 | **$2.2 \times 10^{-6}$** | 1.20 | $4.4 \times 10^{-4}$ |
| ERCC1 | ENSG00000012061 | 19q13.32 | Ovary | 1.19 | **$2.8 \times 10^{-7}$** | 1.24 | $1.8 \times 10^{-4}$ |
| RELB | ENSG00000104856 | 19q13.32 | Lung | 0.57 | **$2.0 \times 10^{-7}$** | 0.44 | $2.9 \times 10^{-6}$ |
| DMPK | ENSG00000104936 | 19q13.32 | Stomach | 1.64 | **$1.7 \times 10^{-6}$** | 2.31 | $1.8 \times 10^{-6}$ |
| CD3EAP | ENSG00000117877 | 19q13.32 | Brain substantia nigra | 0.51 | **$8.0 \times 10^{-17}$** | 0.36 | **$3.8 \times 10^{-15}$** |
| PVRL2 | ENSG00000130202 | 19q13.32 | Artery coronary | 1.36 | **$5.0 \times 10^{-7}$** | 1.59 | $1.6 \times 10^{-6}$ |
| PVRL2 | ENSG00000130202 | 19q13.32 | Oesophagus muscularis | 1.62 | **$6.6 \times 10^{-7}$** | 2.31 | **$4.4 \times 10^{-8}$** |
| GEMIN7 | ENSG00000142252 | 19q13.32 | Brain nucleus accumbens basal ganglia | 0.85 | $1.5 \times 10^{-4}$ | 0.70 | **$1.4 \times 10^{-7}$** |
| BLOC1S3 | ENSG00000189114 | 19q13.32 | Oesophagus muscularis | 2.80 | **$6.4 \times 10^{-16}$** | 4.47 | **$1.3 \times 10^{-13}$** |
| APOC2 | ENSG00000234906 | 19q13.32 | Skin not sun exposed suprapubic | 0.75 | **$4.2 \times 10^{-7}$** | 0.74 | $9.3 \times 10^{-4}$ |

OR odds ratio (i.e., odds to become long-lived when having an increased tissue-specific gene expression). P-values highlighted in bold are significant after adjustment for multiple testing of 247,999 longevity associations with gene-tissue pairs (Storey q-value < 0.05). $OR_{90}$ and $P_{90}$ are based on the analysis of the 90th percentile cases versus all controls meta-analysis data set, while $OR_{99}$ and $P_{99}$ are based on the analysis of the 99th percentile cases versus all controls meta-analysis data set

**Table 6 Results of the genetic correlation analyses of the 90th and 99th percentile phenotypes with other diseases and traits**

| Disease/trait | $rg_{90}$ | $SE_{90}$ | $P_{90}$ | $rg_{99}$ | $SE_{99}$ | $P_{99}$ |
|---|---|---|---|---|---|---|
| Coronary artery disease | −0.40 | 0.07 | **$1.7 \times 10^{-8}$** | −0.29 | 0.07 | **$1.2 \times 10^{-5}$** |
| Fathers age at death | 0.74 | 0.13 | **$2.5 \times 10^{-8}$** | 0.54 | 0.13 | **$2.7 \times 10^{-5}$** |
| HDL cholesterol | 0.36 | 0.07 | **$1.0 \times 10^{-7}$** | 0.22 | 0.07 | 0.002 |
| Age of first birth | 0.33 | 0.07 | **$3.8 \times 10^{-7}$** | 0.16 | 0.07 | 0.019 |
| Years of schooling 2016 | 0.26 | 0.05 | **$9.6 \times 10^{-7}$** | 0.12 | 0.05 | 0.017 |
| Waist circumference | −0.26 | 0.05 | **$2.4 \times 10^{-6}$** | −0.19 | 0.06 | 0.001 |
| Type 2 diabetes | −0.44 | 0.10 | **$4.4 \times 10^{-6}$** | −0.42 | 0.10 | **$2.0 \times 10^{-5}$** |
| Overweight | −0.28 | 0.06 | **$1.2 \times 10^{-5}$** | −0.23 | 0.07 | $9.0 \times 10^{-4}$ |
| Fasting insulin main effect | −0.45 | 0.11 | **$3.0 \times 10^{-5}$** | −0.33 | 0.11 | 0.002 |
| Urate | −0.26 | 0.07 | **$5.0 \times 10^{-5}$** | −0.15 | 0.06 | 0.013 |
| Body mass index | −0.21 | 0.05 | **$9.2 \times 10^{-5}$** | −0.19 | 0.07 | 0.004 |
| Cigarettes smoked per day | −0.49 | 0.13 | **$1.0 \times 10^{-4}$** | −0.31 | 0.13 | 0.016 |
| Mothers age at death | 0.51 | 0.14 | **$2.0 \times 10^{-4}$** | 0.14 | 0.13 | 0.289 |
| Waist-to-hip ratio | −0.24 | 0.07 | **$2.0 \times 10^{-4}$** | −0.15 | 0.07 | 0.028 |

P-values highlighted in bold are significant after Bonferroni adjustment for multiple testing (P < 0.05/246). $rg_{90}$, $SE_{90}$, and $P_{90}$ are based on the analysis of the 90th percentile cases versus all controls meta-analysis data set, while $rg_{99}$, $SE_{99}$, and $P_{99}$ are based on the analysis of the 99th percentile cases versus all controls meta-analysis data set
rg genetic correlation, SE standard error of the rg estimate, HDL high-density lipoprotein

diseases, such as cardiovascular disease and Alzheimer's disease[33], which could explain their effect on longevity. The fact that the two variants in ApoE show opposite effects may be attributable to differences in structural and biophysical properties of the protein, since ApoE ε2 shows high stability and ApoE ε4 low stability upon folding[34].

We also found a genome-wide significant association of rs7676745, located on chromosome 4 near GPR78. We have to note that this locus would benefit from replication in independent cohorts in the future, given that we were not able to replicate this variant in the cohorts in which de novo genotyping was applied. There is no report of association of this locus with other traits according to Phenoscanner (http://www.phenoscanner.medschl.cam.ac.uk/)[35], although other genetic variants in this gene have been associated with several diseases and traits in the UK Biobank, including death due to a variety of disorders. The GPR78 protein, belongs to the family of G-protein-coupled receptors, whose main function is to mediate physiological responses to

various extracellular signals, including hormones and neurotransmitters[36]. However, the specific function of GPR78 is still largely unknown, although it has been shown to play a role in lung cancer metastasis[37].

To maximise power for discovery, we meta-analysed results from all of the studies that contained long-lived individuals that met our 90th and/or 99th percentile case definitions, had genome-wide genetic data, and were able to participate. Hence, we were not able to replicate our findings in an independent cohort with genome-wide genotype data and participants reaching the age of our case definitions. Therefore, we tried to validate our findings using two related phenotypes, parental longevity and lifespan, in the UK Biobank. We applied our case and control definitions to the parental lifespan of genotyped middle-aged UK Biobank participants rather than the participants themselves, as none of the latter fulfilled the age criteria for cases in our study. Although this resulted in relatively large data sets for both the 90th and 99th percentile analysis, the power to replicate our

findings using the parental longevity traits was lower in comparison to replication using the traits based on the genotyped individuals themselves, since these individuals share only half of their parental genomes. In addition, many of the genotyped individuals, who were 40–69 years at recruitment, will never reach the age belonging to the 90th, let alone the 99th, percentile of their birth cohort. This may explain why we were unable to validate any of our suggestive associations ($P \leq 1 \times 10^{-6}$), with the exception of the genetic variants at the *APOE* locus in these data sets. On the other hand, we were able to validate one additional locus, *CDKN2A/B*, in the parental lifespan data set. This is not surprising, since this locus had already been reported to associate with parental lifespan[20]. However, it is unclear why our reported variants at this locus, rs7039467 and rs2184061, are not associated with parental longevity, given that the most significant parental lifespan-associated variant at this locus, rs1556516, also shows a nominal significant effect on parental longevity (see Supplementary Table 2). We hypothesise that this may be due to a difference in the LD structure of the reference panels used for imputation.

We were able to detect significant genetic associations at two previously identified longevity/lifespan-related loci, *FOXO3* and *CDKN2A/B*. For the other loci, we did not find evidence for replication ($P > 7.8 \times 10^{-4}$), despite having adequate power ($\geq 0.8$) for replication of all but one of the examined genetic variants (rs28926173) associated with the discrete longevity phenotypes. We were not able to calculate our power to replicate the variants associated with the continuous lifespan-related phenotypes, although we should have had adequate power to replicate variants with a minor allele frequency (MAF) > 12% and an OR > 1.1 (based on the 90th percentile versus all controls analysis). However, several of the variants associated with parental lifespan show a directionally consistent and nominal significant association with our phenotypes, indicating they may also be relevant for longevity. The failure to replicate previously reported loci could be due to the use of a different longevity phenotype then what was used in previous studies, the small effect size of some of the variants associated with parental lifespan, and the modest power of our study. The fact that we detect significant associations of variants in the *FOXO3* locus is not surprising, since this locus was previously reported in the longevity GWA study from the CHARGE consortium[7], from which many cohorts are included in these meta-analyses. So far, three functional longevity-associated variants have been identified at the *FOXO3* locus (rs2802292, rs12206094, and rs4946935). For all of them, an allele-specific response to cellular stress was observed. Consistently, the longevity-associated alleles of all three variants were shown to induce *FOXO3* expression[38,39]. The *CDKN2A/B* locus has previously been associated with parental lifespan and parents' attained age in the UK Biobank as well as a diversity of age-related diseases[13,20,40]. The longevity-associated allele of the most significant variant at this locus (rs1556516) has also been associated with lower odds of developing CAD[41]. Although the molecular mechanism behind this association is still unclear, it is known that genes encoded at the *CDKN2A/B* locus are involved in cellular senescence[42], a known hallmark of ageing in animal models[43].

The gene-level association analysis identified several associations between increased (*KANSL1*, *CRHR1*, *ARL17A*, and *LRRC37A2*) or decreased (*ANKRD31* and *BLOC1S1*) genetically driven tissue-specific gene expression with survival to the 90th percentile age. The increased expression of *KANSL1*, *CRHR1*, *ARL17A*, and *LRRC37A2* on chromosome 17q21.31 is regulated by different genetic variants, indicating that these associations may be independent. More functional work is needed to determine the exact relationship between the altered genetically driven tissue-specific expression of these genes and longevity in humans.

A limitation of MetaXcan is that the underlying GTEx models might not have been adequately adjusted for age, which could be problematic for an age-related phenotype like longevity. However, MetaXcan has successfully been used to identify gene-level associations with age-related diseases and traits, such as Alzheimer's disease and age-related macular degeneration[25].

The genetic correlation analyses showed that survival to ages corresponding to the 90th and 99th percentile shared genetic associations with father's age at death, CAD and T2D-related phenotypes, suggesting that survival to old ages may at least partially be explained by protective influences on the mechanisms underlying these traits. The genetic correlation with CAD and T2D-related phenotypes is expected, since it has previously been reported that individuals from long-lived families show a decreased prevalence of cardiovascular disease and T2D[44,45]. The higher genetic correlation of our longevity phenotypes with father's in comparison to mother's age at death may be explained by the difference in the prevalence of cardiovascular diseases and T2D between men and women in the last century[46,47], which may be, at least partially, attributable to a difference in smoking prevalence[48]. Hence, the correlation of our longevity phenotypes with the parental age at death phenotypes from UK Biobank is likely due to the absence of death from specific diseases (i.e., those with a higher prevalence in men). For longevity-specific loci, on the other hand, one would expect that they will have beneficial effects on multiple diseases simultaneously, since long-lived individuals show a delay in overall morbidity[49].

Our study design imposed an age gap between cases and controls to reduce outcome misclassification, which we expected could potentially increase power by increasing the genetic effect size. It has been correctly noted that longevity study designs that include an age gap between cases and controls result in an effect estimate that is based on an OR and a relative risk (RR) term, which could lead to the identification of genetic variant associations related to early mortality (OR), rather than survival past the case age threshold (RR) (for more details see Sebastiani et al.)[50]. However, we have presented evidence that imposing a case–control age gap did not greatly influence our results or prevent our replication of variant associations previously discovered using study designs without a case–control age gap. First, our sensitivity analysis indicated that reducing the age gap between cases and controls had a minimal effect on our results. Our sensitivity analysis compared results using dead controls, where all individuals had died before they reached the 60th percentile age, and all controls, which included dead controls and individuals whose age at last contact was below the 60th percentile age but whose age of death was unknown. There is likely to be some outcome misclassification of the living controls, since a small percentage may survive beyond the age corresponding to the 90th or 99th survival percentile. On the other hand, the age gap between cases and controls was narrower for all controls compared to dead controls. However, despite the narrower age gap, the suggestively significant results in all controls and dead controls comparisons with 90th percentile cases were essentially unchanged, and there was a very high genetic correlation between the results of these two meta-analyses, indicating that the age gap had little or no impact on our results. Second, if we had discovered a large number of genome-wide significant variant associations in our study, it could be argued that the OR, reflecting early mortality, contributed to some or all of them. However, the only genome-wide significant variant associations we detected were in the *APOE* locus, which have been identified using multiple study designs, including designs with no pre-specified age gap between cases and controls[14], and the *GPR78* locus. Third, it is unlikely that our study design prevented the replication of findings from previous GWA studies of survival to

extreme ages (i.e., 99th percentile cases) that did not include a case–control age gap, since such studies would only identify variants associated with survival past the minimum case age and not with early mortality. For variants with no early mortality association, it would be expected that the association estimate in our study would have an OR equal to one and a RR greater than one. Nothing prevents our study design from also detecting this type of variant association, as our estimated association parameter reflects both the OR and RR.

The majority of the previously performed GWA studies of longevity used the survival of individuals to a pre-defined age threshold (i.e., 85, 90, or 100 years) as selection criterion to define long-lived cases. Although these studies used a consistent phenotype for each cohort included in the GWA study, this type of selection may gave rise to heterogeneity, given that survival probabilities differ between sexes and birth cohorts[22]. Moreover, it was recently shown that the heritable component of longevity is strongest in individuals belonging to the top 10% survivors of their birth cohort[6]. Hence, instead of using a pre-defined age threshold, we decided to select cases based on country-, sex- and birth cohort-specific life tables. For the definition of controls we used the 60th percentile age, since we wanted to include as many controls as possible (preferably from the same cohort as the cases), while leaving a large enough age gap between our cases and controls. Using the 1920 birth cohort as an example, the difference between the 60th and 90th percentile age is 14 years (men) or 11 years (women), which is quite substantial. The difference between the 70th and 90th percentile age, on the other hand, is considerably smaller (9 years (men) or 7 years (women)) and the living controls are more likely to reach the 90th percentile age, which increases the risk of outcome misclassification. Moreover, even when selecting the 60th percentile controls from much later birth cohorts (i.e., 1940) than the cases (i.e., 1900) the ages will not overlap.

Our study has several limitations. First, we did not analyse the sex and mitochondrial chromosomes, since we were unable to gather enough cohorts that could contribute to the analysis of these chromosomes. However, these chromosomes may harbour loci associated with longevity that we thus have missed. Second, although we included as many cohorts as possible, the sample size of our study is still relatively small (especially for the 99th percentile analysis) in comparison to GWA studies of age-related diseases, such as T2D and cardiovascular disease, and parental age at death[11,51,52]. Hence, this limited our power to detect loci with a low MAF (<1%) that contribute to longevity. Third, we did not perform sex-stratified analyses and may thus have missed sex-specific longevity-related genetic variants. The reason for this is that (1) we only identified a limited number of suggestive significant associations in our unstratified 90th and 99th percentile analyses, (2) our sample size is modest (especially when stratified by sex), and (3) thus far, there has been no report of any genome-wide significant sex-specific longevity locus.

Given that we have included nearly all cohorts with long-lived individuals with genome-wide genetic data in our study, it will be challenging to increase the sample size in future GWA studies using the same extreme phenotypes. Future genetic studies of longevity may therefore benefit from the use of alternative phenotypes or more rigorous phenotype definitions. Alternative phenotypes that could be used are the parental lifespan or healthspan-related phenotypes that were analysed in the UK Biobank or biomarkers of healthy aging[20,53,54]. One way to strengthen the longevity phenotype is by selecting cases from families with multiple individuals belonging to the top 10% survivors of their birth cohort[6]. Moreover, given the limited number of longevity-associated genetic variants identified through GWA studies and the availability of affordable exome and whole-

genome sequencing, future genetic studies of longevity may also benefit from the analysis of rare genetic variants. Ideally, such studies should also try to include participants from genetically diverse populations. Most cohorts that are currently included in genetic longevity studies originate from populations of European descent, while some longevity loci may be specific for non-European populations, as exemplified by the previously reported genome-wide associations of genetic variants in IL6 and ANKRD20A9P in Han Chinese[9]. Moreover, a recent genetic study of multiple complex traits has shown the benefit of analysis of diverse populations[55].

In conclusion, we performed a genome-wide association study of longevity-related phenotypes in individuals of European, East Asian and African American ancestry and identified the APOE and GPR78 loci to be associated with these phenotypes in our study. Moreover, our gene-level association analyses highlight a role for tissue-specific expression of genes at chromosome 5q13.3, 12q13.2, 17q21.31, and 19q13.32 in longevity. Genetic correlation analyses show that our longevity-related phenotypes are genetically correlated with several disease-related phenotypes, which in turn could help to identify phenotypes that could be used as potential biomarkers for longevity in future (genetic) studies.

## Methods

**Study populations.** In this collaborative effort, we included cohorts that participated in one or more of the previously published GWA studies on longevity[7–9]. The sample sizes and descriptive characteristics of the cohorts used in this study are provided in Table 1, Supplementary Data 4, and the Supplementary Methods.

We have complied with all relevant ethical regulations for work with human subjects. All participants provided written informed consent and the studies were approved by the relevant institutional review boards.

**Case and control definitions.** Cases were individuals who lived to an age above the 90th or 99th percentile based on cohort life tables from census data from the appropriate country, sex, and birth cohort. Controls were individuals who died at or before the age at the 60th percentile or whose age at the last follow-up visit was at or before the 60th percentile age. Hence, the number of selected cases and controls is defined by the ages of their birth cohort corresponding to the 60th or 90th/99th percentile age and is independent of the study population used (i.e., the number of controls and cases within a study population is not based on the percentiles of that specific population, but instead on that of their birth cohorts). As part of their recruitment protocol, many of the studies enroled participants that were already relatively old at the time of recruitment (i.e., close to (or even over) the 60th percentile age). The majority of these individuals subsequently survived past the 60th percentile age threshold of their respective birth cohorts, resulting in a small number of controls in comparison to the number of cases for some of these studies.

The cohort life tables were available through the Human Mortality Database (www.mortality.org)[56], the United States Social Security Administration (https://www.ssa.gov/oact/NOTES/as120/LifeTables_Tbl_7.html)[22] or National registries; https://opendata.cbs.nl/statline/portal.html?_la = nl&_catalog = CBS&tableId = 80333ned&_theme = 90; http://webarchive.nationalarchives.gov.uk/20160129121820/http://www.ons.gov.uk/ons/rel/npp/national-population-projections/2012-based-extra-variants/index.html) 2012. For example, the 60th, 90th, and 99th percentile correspond to ages of 75, 89, and 98 years for men and 83, 94, and 102 years for women for the 1920 birth cohort from the US. For cohort life tables providing birth cohort by decade, linear model predictions were used to estimate the ages corresponding to survival percentiles at yearly birth cohorts.

For the parental longevity analyses in the UK Biobank, cases were individuals with at least one parent achieving an age above the 90th or 99th percentile and who had not themselves died, while controls were individuals for whom both parents died at or before the age at the 60th percentile.

**Genome-wide association analysis of individual cohorts.** Details on the genotyping (platform and quality control criteria), imputation and genome-wide association analyses for each cohort are provided in Supplementary Data 5. In all cohorts, genetic variants were imputed using the 1000G Phase 1 version 3 reference panel. The logistic regression analyses were adjusted for clinical site, known family relationships, and/or the first four principal components (if applicable). All cohorts used a Hardy–Weinberg equilibrium (HWE) P-value that was between $1 \times 10^{-4}$ and $1 \times 10^{-6}$ to exclude variants not in HWE, which is considered standard in GWA studies. However, this may have resulted in removal of variants that were out of HWE in the cases due to mortality selection[57].

**Quality control of individual cohorts**. Quality control of the summary statistics from each cohort was performed using the EasyQC software and the standard script (fileqc_1000G.ecf) available on their website (http://www.uni-regensburg.de/medizin/epidemiologie-praeventivmedizin/genetische-epidemiologie/software/)[58]. The only difference was that we used the expected minor allele count (eMAC) instead of the MAC. To this end, we first calculated the 'Effective $N$' ($2/(1/N_{cases} + 1/N_{controls})$) for each cohort. The use of the 'Effective $N$' instead of the 'Total $N$' leads to a more stringent filtering of genetic variants and decreases the chance of false positive findings due to an imbalance between the number of cases and controls[58]. The 'Effective $N$' was subsequently used to calculate the eMAC (2 × minor allele frequency × 'Effective $N$' × imputation quality) for each variant. Variants were excluded when eMAC < 10, with the exception of the Newcastle 85 + (90th percentile cases versus all controls) and the RS (99th percentile cases versus all controls) data sets in which we excluded variants when eMAC < 25 due to the large imbalance between the number of cases and controls in these data sets (1:24 and 1:38, respectively) in comparison to the other ones (all < 1:10). For the CLHLS and LLFS data sets, we flipped the strands of several variants based on the discordance of allele frequencies with the reference panel. We only flipped palindromic variants with a MAF < 0.4 and an allele frequency that differed from the reference panel by <10% after switching.

**Meta-analyses**. The fixed-effect meta-analyses based on the data sets with individuals of European ancestry were performed on the cleaned files using METAL[59], with the 'Effective $N$' as weight and adjustment for genomic control (lambda ($\lambda$)) for each cohort. Cohorts with an 'Effective $N$' < 50 were excluded from the meta-analyses. We did not apply genomic control on the meta-analyses results, since there was limited inflation (all $\lambda$ < 1.04, Supplementary Fig. 1).

The trans-ethnic meta-analyses were performed using the random-effects model of Han and Eskin, implemented in METASOFT[60]. This model separates hypothesis testing from the estimation of the effect size, which allows the test to better model the between-study heterogeneity that is typically encountered in a trans-ethnic meta-analysis. Prior to using METASOFT, study-specific results were filtered as described above, which included removing genetic variants with eMAC < 10, and applying genomic control by multiplying each variant's standard error by the inverse of the square root of the lambda for cohorts with $\lambda$ > 1.

Genetic variants for which the total 'Effective $N$' was less than half of the maximum 'Effective $N$' were removed from the meta-analyses results.

**Conditional analyses**. Conditional analyses were performed using the '-condition_on' option implemented in SNPTEST to determine the number of independent signals at the *APOE* locus. We performed this analysis in the cohorts that were analysed using SNPTEST and for which both the ApoE ε4 and ApoE ε2 variant showed a significant association in the unadjusted analysis (i.e., CEPH and LLS (combined with GEHA Dutch)). In both cohorts, the association of ApoE ε2 remained significant ($P$ < 0.05) after adjustment for ApoE ε4, indicating an independent effect.

**Gene-level association analysis**. MetaXcan was used to identify genetically predicted tissue-specific expression associations with longevity using the results from the 90th and 99th percentile cases versus all controls meta-analyses[25]. GTEx version 7 tissue models of genetically predicted expression were used. To maximize the number of genetic variants that MetaXcan could match with tissue models, the MetaXcan SNP annotation file (gtex_v7_hapmapceu_dbsnp150_snp_annot.txt) was used to map variants from the GWA study results file to rsIDs by chromosome, position, and alleles. To control for the false discovery rate when testing multiple genes across multiple tissues, the Storey $q$-value was applied and a $q$-value < 0.05 was considered significant[61].

Colocalization of the tissue-specific eQTL results from the GTEx project and our longevity meta-analyses results was performed using the 'coloc.abf' function implemented in the R-package coloc[62].

**Genetic correlation analysis**. To estimate the genetic correlation between the different phenotypes used in this study, we used LD score regression[24]. The genetic correlation between the results from the 90th and 99th percentile cases versus all controls meta-analyses and 246 diseases and traits were estimated using the LD Hub web portal (http://ldsc.broadinstitute.org/ldhub/)[26]. Since LD score regression is currently only possible with data from individuals of European ancestry, we used our meta-analyses results based on the cohorts from populations of European descent only.

**Power calculation**. The power calculations for the validation in the UK Biobank and for the replication of previously identified loci associated with human lifespan were performed using the Genetic Association Study Power Calculator (http://csg.sph.umich.edu/abecasis/cats/gas_power_calculator/index.html) using an additive disease model and a disease prevalence of 0.1 (90th percentile) or 0.01 (99th percentile).

**Reporting summary**. Further information on research design is available in the Nature Research Reporting Summary linked to this article.

## Data availability

The full meta-analyses summary statistics are available for download at www.longevitygenomics.org/downloads and through GRASP (https://grasp.nhlbi.nih.gov/FullResults.aspx) and the NHGRI-EBI GWAS Catalog (https://www.ebi.ac.uk/gwas/downloads/summary-statistics). All other data that supports the findings of this study are available from the corresponding authors upon request.

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

## Acknowledgements

A full list of acknowledgements, including support for each of the participating cohorts, is provided in Supplementary Note 1.

## Author contributions

J.D., D.S.E., P.E.S., and J.M.M. designed and supervised the study. J.D. and D.S.E. performed the meta-analyses and follow-up experiments. J.D., D.S.E., D.P.K., K.L., P.E.S., and J.M.M. interpreted the data. J.D., D.S.E., D.E.A., N.T., M.N., X.L., M.K.W., M.L.B., A. vdS., G.A., E.B.W., C.S., A.V.S., I.S., H.J.C., J.Do., N.A., A.M.A., K.L.A., E.J.B., B.D.,C.J., A.K., L.-P.L., J.M., P.S., S.J.vdL., K.Y., and W.Z. performed association analyses for individual cohorts. G.A., M.B., J.D.F., E.S.O., J.Do., H.B., J-F.D., P.G., C.M.-R., C.N., J.I. R., J.S., K.D.T., A.G.U., J.C.C., S.C., S.R.C., K.D., V.G., D.K., W.L., A.B.N., M.J.T.R., J.-M. R., D.vH., and J.W.V. were involved in data acquisition and/or genotyping of the individual cohorts. N.B., H.B., K.C., L.C., S.R.C., J.-F.D., R.D., C.F., P.G., V.G., T.B.H., M.H., M.A.H., C.J., I.J., M.J., M.K., S.L.R.K., T.B.L.K., L.J.L., T.L., A.N., E.A.N., E.S.O., T.T.P., M.A.P., B.M.P., O.T.R., T.I.A.S., W.vdF., C.M.vD., D.W., Y.Z., H.H., D.P.K., K.L., P.E.S., and J.M.M. were involved in supervision of individual cohorts. J.D., D.S.E., P.E.S., and J.M.M. wrote the manuscript. All authors read and approved the final version of the manuscript.

## Additional information

**Competing interests:** The authors declare no competing interests.

Joris Deelen[1,2,78], Daniel S. Evans[3,78], Dan E. Arking[4], Niccolò Tesi[5,6,7], Marianne Nygaard[8], Xiaomin Liu[9,10], Mary K. Wojczynski[11], Mary L. Biggs[12,13], Ashley van der Spek[14], Gil Atzmon[15,16], Erin B. Ware[17], Chloé Sarnowski[18], Albert V. Smith[19,20], Ilkka Seppälä[21], Heather J. Cordell[22], Janina Dose[23], Najaf Amin[14], Alice M. Arnold[12], Kristin L. Ayers[24], Nir Barzilai[16], Elizabeth J. Becker[25], Marian Beekman[2], Hélène Blanché[26], Kaare Christensen[8,27,28], Lene Christiansen[8,29], Joanna C. Collerton[30], Sarah Cubaynes[31], Steven R. Cummings[3], Karen Davies[32], Birgit Debrabant[33], Jean-François Deleuze[26,34], Rachel Duncan[30,35], Jessica D. Faul[17], Claudio Franceschi[36,37], Pilar Galan[38], Vilmundur Gudnason[20,39], Tamara B. Harris[40], Martijn Huisman[41,42], Mikko A. Hurme[43], Carol Jagger[30,35], Iris Jansen[5,44], Marja Jylhä[45], Mika Kähönen[46], David Karasik[47,48], Sharon L.R. Kardia[49], Andrew Kingston[30,35], Thomas B.L. Kirkwood[35], Lenore J. Launer[40], Terho Lehtimäki[21], Wolfgang Lieb[50], Leo-Pekka Lyytikäinen[21], Carmen Martin-Ruiz[32], Junxia Min[51], Almut Nebel[23], Anne B. Newman[52], Chao Nie[9], Ellen A. Nohr[53], Eric S. Orwoll[54], Thomas T. Perls[55], Michael A. Province[11], Bruce M. Psaty[13,56,57,58], Olli T. Raitakari[59,60], Marcel J.T. Reinders[7], Jean-Marie Robine[31,61], Jerome I. Rotter[62,63], Paola Sebastiani[18], Jennifer Smith[17,49], Thorkild I.A. Sørensen[64,65], Kent D. Taylor[62,66], André G. Uitterlinden[14,67], Wiesje van der Flier[5,41], Sven J. van der Lee[5,6], Cornelia M. van Duijn[14,68], Diana van Heemst[69], James W. Vaupel[70], David Weir[17], Kenny Ye[71], Yi Zeng[72,73], Wanlin Zheng[3], Henne Holstege[5,6,7], Douglas P. Kiel[48,74,75], Kathryn L. Lunetta[18], P. Eline Slagboom[2,78] & Joanne M. Murabito[76,77,78]

[1]Max Planck Institute for Biology of Ageing, 50866 Cologne, Germany. [2]Molecular Epidemiology, Department of Biomedical Data Sciences, Leiden University Medical Center, 2300 RC Leiden, The Netherlands. [3]California Pacific Medical Center Research Institute, San Francisco, CA 94158, USA. [4]McKusick-Nathans Institute of Genetic Medicine, Department of Genetic Medicine, Johns Hopkins University School of Medicine, Baltimore, MD 21287, USA. [5]Alzheimer Center Amsterdam, Department of Neurology, Amsterdam Neuroscience, Vrije Universiteit Amsterdam, Amsterdam UMC, 1007 MB Amsterdam, The Netherlands. [6]Department of Clinical Genetics, Amsterdam UMC, 1007 MB Amsterdam, The Netherlands. [7]Delft Bioinformatics Lab, Delft University of Technology, 2600 GA Delft, The Netherlands. [8]The Danish Aging Research Center, Department of Public Health, University of Southern Denmark, 5000 Odense C, Denmark. [9]BGI-Shenzhen, Shenzhen 518083, China. [10]China National Genebank, BGI-Shenzhen, Shenzhen 518120, China. [11]Division of Statistical Genomics, Department of Genetics, Washington University School of Medicine, Saint Louis, MO 63110, USA. [12]Department of Biostatistics, University of Washington, Seattle, WA 98115, USA. [13]Cardiovascular Health Research Unit, Department of Medicine, University of Washington, Seattle, WA 98101, USA. [14]Department of Epidemiology, Erasmus MC, 3000 CA Rotterdam, The Netherlands. [15]Department of Biology, Faculty of Natural Science, University of Haifa, Haifa 3498838, Israel. [16]Departments of Medicine and Genetics, Albert Einstein College of Medicine, Bronx, NY 10461, USA. [17]Institute for Social Research, Survey Research Center, University of Michigan, Ann Arbor, MI 48104, USA. [18]Department of Biostatistics, Boston University School of Public Health, Boston, MA 02118, USA. [19]School of Public Health, Department of Biostatistics, University of Michigan, Ann Arbor, MI 48109, USA. [20]Icelandic Heart Association, 201 Kópavogur, Iceland. [21]Department of Clinical Chemistry, Fimlab Laboratories and Finnish Cardiovascular Research Center—Tampere, Faculty of Medicine and Health Technology, Tampere University, 33520 Tampere, Finland. [22]Institute of Genetic Medicine, Newcastle University, Newcastle upon Tyne NE1 3BZ, UK. [23]Institute of Clinical Molecular Biology, Kiel University, 24105 Kiel, Germany. [24]Sema4, a Mount Sinai venture, Stamford, CT 06902, USA. [25]Bioinformatics Program, Boston University, Boston, MA 02118, USA. [26]Fondation Jean Dausset—CEPH, 75010 Paris, France. [27]Clinical Biochemistry and Pharmacology, Odense University Hospital, 5000 Odense C, Denmark. [28]Department of Clinical Genetics, Odense University Hospital, 5000 Odense C, Denmark. [29]Department of Clinical Immunology, Copenhagen University Hospital, Rigshospitalet, 2100 Copenhagen, Denmark. [30]Institute of Health & Society, Newcastle University, Newcastle upon Tyne NE4 5PL, UK. [31]MMDN, Univ. Montpellier, EPHE, Unité Inserm 1198, PSL Research University, 34095 Montpellier, France. [32]Institute of Neuroscience, Newcastle University, Newcastle upon Tyne NE4 5PL, UK. [33]Department of Public Health, University of Southern Denmark, 5000 Odense C, Denmark. [34]Centre National de Recherche en Génomique Humaine, CEA-Institut de Biologie François Jacob, 91000 Evry, France. [35]Newcastle University Institute for Ageing, Newcastle University, Newcastle upon Tyne NE4 5PL, UK. [36]Department of Applied Mathematics and Centre of Bioinformatics, Lobachevsky State University of Nizhny Novgorod, Nizhny Novgorod 603022, Russia. [37]IRCCS Institute of Neurological Sciences of Bologna (ISNB), 40124 Bologna, Italy. [38]EREN, UMR U1153 Inserm/U1125 Inra/Cnam/Paris 13, Université Paris 13, CRESS, 93017 Bobigny, France. [39]Faculty of Medicine, University of Iceland, 101 Reykjavik, Iceland. [40]Laboratory of Epidemiology and Population Sciences, National Institute on Aging, NIH, Bethesda, MD 20892, USA. [41]Department of Epidemiology and Biostatistics, Vrije Universiteit Amsterdam, Amsterdam UMC, 1007 MB Amsterdam, The Netherlands. [42]Amsterdam Public Health Research Institute, 1007 MB Amsterdam, The Netherlands. [43]Department of Microbiology and Immunology, Faculty of Medicine and Health Technology, Tampere University, 33014 Tampere, Finland. [44]Department of Complex Trait Genetics, Center for Neurogenomics and Cognitive Research, Vrije Universiteit Amsterdam, 1081 HV Amsterdam, The Netherlands. [45]Faculty of Social Sciences (Health Sciences) and Gerontology Research Center (GEREC), Tampere University, 33104 Tampere, Finland. [46]Department of Clinical Physiology, Tampere University Hospital and Finnish Cardiovascular Research Center—Tampere, Faculty of Medicine and Health Technology, Tampere University, 33521 Tampere, Finland. [47]Azrieli Faculty of Medicine, Bar Ilan University, Safed 13010, Israel. [48]Hinda and Arthur Marcus Institute for Aging Research, Hebrew SeniorLife, Boston, MA 02131, USA. [49]School of Public Health, Epidemiology, University of Michigan, Ann Arbor, MI 48109, USA. [50]Institute of Epidemiology and Biobank PopGen, Kiel University, 24105 Kiel, Germany. [51]Institute of Translational Medicine, School of Medicine, Zhejiang University, Hangzhou 311058, China. [52]Department of Epidemiology, Graduate School of Public Health, University of Pittsburgh, Pittsburgh,

PA 15261, USA. [53]Research Unit of Gynecology and Obstetrics, Department of Clinical Research, University of Southern Denmark, 5000 Odense C, Denmark. [54]Bone and Mineral Unit, Oregon Health Sciences University, Portland, OR 97239, USA. [55]Department of Medicine, Geriatrics Section, Boston Medical Center, Boston University School of Medicine, Boston, MA 02118, USA. [56]Department of Epidemiology, University of Washington, Seattle, WA 98101, USA. [57]Department of Health Services, University of Washington, Seattle, WA 98101, USA. [58]Kaiser Permanente Washington Health Research Institute, Seattle, WA 98101, USA. [59]Department of Clinical Physiology and Nuclear Medicine, Turku University Hospital, 20521 Turku, Finland. [60]Research Centre of Applied and Preventive Cardiovascular Medicine, University of Turku, 20014 Turku, Finland. [61]CERMES3, UMR CNRS 8211—Unité Inserm 988—EHESS—Université Paris Descartes, 94801 Paris, France. [62]Institute for Translational Genomics and Population Sciences, Los Angeles Biomedical Research Institute at Harbor-UCLA Medical Center, Torrance, CA 90502, USA. [63]Division of Genetic Outcomes, Department of Pediatrics, Harbor-UCLA Medical Center, Torrance, CA 90502, USA. [64]Novo Nordisk Foundation Center for Basic Metabolic Research, Section of Metabolic Genetics, and Department of Public Health, Section of Epidemiology, Faculty of Health and Medical Sciences, University of Copenhagen, 2200 Copenhagen N, Denmark. [65]MRC Integrative Epidemiology Unit, Bristol University, BS8 2BN Bristol, UK. [66]Department of Pediatrics, Harbor-UCLA Medical Center, Torrance, CA 90502, USA. [67]Department of Internal Medicine, Erasmus MC, 3000 CA Rotterdam, The Netherlands. [68]Nuffield Department of Population Health, University of Oxford, Oxford OX3 7LF, UK. [69]Department of Gerontology and Geriatrics, Leiden University Medical Center, 2300 RC Leiden, The Netherlands. [70]Max Planck Institute for Demographic Research, 18057 Rostock, Germany. [71]Department of Epidemiology and Population Health, Albert Einstein College of Medicine, Bronx, NY 10461, USA. [72]Center for Healthy Aging and Development Studies, National School of Development and Raissun Institute for Advanced Studies, Peking University, 100871 Beijing, China. [73]Center for the Study of Aging and Human Development and Geriatrics Division, Medical School of Duke University, Durham, NC 27710, USA. [74]Department of Medicine, Beth Israel Deaconess Medical Center and Harvard Medical School, Boston, MA 02215, USA. [75]Broad Institute of MIT & Harvard, Cambridge, MA 02142, USA. [76]NHLBI's and Boston University's Framingham Heart Study, Framingham, MA 01702, USA. [77]Section of General Internal Medicine, Department of Medicine, Boston University School of Medicine, Boston, MA 02118, USA. [78]These authors contributed equally: Joris Deelen, Daniel S. Evans, P. Eline Slagboom, Joanne M. Murabito.

