## [Peer Review File · Nature Communications]

Reviewers' Comments:

Reviewer #1:

Remarks to the Author:

In a clean and tidy meta-analysis of longevity, looking at 90th and 99th percentiles of survivorship within each cohort, the authors confirm at unprecedented levels of statistical significance, the known deleterious effect of APOE e4 and identify a clear signal for an independent beneficial variant APOE e2 at this locus.

The authors go on to identify an association at GPR78 with lower levels of statistical significance and using gene level analyses to identify associations at CRHR1. They go onto show genetic correlations between longevity and a number of health related traits.

The combination of data using consistent phenotyping is a positive step forward. The statistical methods applied are well tried and appear appropriate in their application.

The identification of GPR78 is interesting. Whilst the authors are commendably transparent in the results used to make the inference, the combined evidence is of medium, rather than great strength. A simpler combined data presentation might make the paper clearer, without losing any robust important results. Robustness could be increased with a much bigger sample, but this is unrealistic.

Comments

Major

line 296: Nature Communications has an admirable policy on data disclosure, it would be helpful to review (a small sample of) the specific fields and field labels of the data proposed to be disclosed. Discovery and replication data should both be provided.

~line 324

The separation into discovery and replication, with a relaxed threshold for discovery is not without precedent, but carries risk of Type 1 error. The authors argue that as only 3 tests of replication were performed for the 90th percentile, the 1% p-value for GPR78 passes Bonferoni adjustment. However, 11 replication tests were performed in total, and even if q-values were used (taking APOE out of consideration and removing duplicates), the full set of replication tests may mean the 5% (q or adjusted p) threshold is not exceeded. The relaxed discovery threshold and the less than conclusive replication suggest the GPR78 finding is interesting, but cannot yet be considered fully robust.

Furthermore, rough meta-analysis of discovery and replication suggest that $OR=0.65$ (0.56-0.75). Whereas the UKBB results the authors present suggest $OR=0.96$ (0.84-1.12). This suggests to me the UKBB results contradict, rather than are merely underpowered. {Doubling effects UKBB for the effect of parent imputation, on the assumption this was not done already. Not doubling would have meant an even stronger contradiction.}

The authors should either (a) be more equivocal about the overall picture for GPR78; or (b) combine discovery and replication and then use UKBB as confirmation (failing at GPR78) (UKBB does appear powered to do the job for a reasonably large set of discovered SNPs, based on the above {UKBB SEs appear similar to discovery}); or (preferably, but more work) meta-analyse (b) with UK Biobank too.

~line 376

Units in ST5 appear confused. For Timers et al, the reported betas appear multiplied by 10. The reported betas were In HR, so $\exp(\text{reported beta})$ would seem the most comparable thing to report. For Pilling et al, the number reported is the effect size for rank normalised martingale residuals. This is not an HR and not comparable and should be reported as an effect size (eg beta) with legend explaining the scale used.

~line 479.

Some of the sense here is reasonable, but the most likely reason is that (as implicitly acknowledged), the authors do not have power to detect the presumable modest longevity effects of these loci, given the multiple testing and the power of their study. Consider the following line of reasoning using line 41 and 43 of table S5. LPA/CDKN2B appear an unambiguous lifespan locus. They also appear to be a longevity loci ($OR=1.13/1.06$ $p=0.012/0.002$), but allowing for multiple testing, the authors are not powered enough to confirm this.

Some discussion that the authors are not able to refute that these could well be real lifespan, and perhaps real (modest) longevity loci, ($OR < 1.1$), that did not replicate due to lack of power and perhaps greater effect on mortality than longevity should be included.

Some discussion as to whether present lifespan studies appear to be more sensitive than present longevity studies in picking up longevity loci should be included (perhaps with the caveat that merely avoiding early death isn't the most biologically interesting form of longevity).

The use of consistent phenotyping is to be commended, but some discussion that there is more than one form of consistency (eg 85+, 95+), and that these might be biologically relevant, would be helpful

Minor

~line 217. It would be nice to see a table or graph of the 90th and 99th percentile ages for all cohorts (survival, curves would be even better)

~line 252 replace "heterogeneity" with "inflation"

Line 327,330 : there is a confusion between variant/allele and polymorphism here. Perhaps just specify the allele at line 327?

Reviewer #2:

Remarks to the Author:

The authors performed two meta-analyses of GWA studies of longevity applying a rigorous definition of cases and controls based on country-, sex-, and birth cohort-specific life tables. The discovery phase included 10,889/3,183 cases surviving at or beyond the age corresponding to the 90th/99th survival percentile, respectively, and 23,212 controls whose age at death or at last contact was at or below the age corresponding to the 60th survival percentile. Consistent with previous reports, rs429358 (ApoE $\epsilon 4$) was associated with lower odds of surviving to the 90th and 99th percentile age.

The authors also detected a genome-wide significant association of rs7412 (ApoE $\epsilon 2$) with higher odds of surviving to the 90th and 99th percentile age. One SNP, rs7676745, located on chromosome 4 near GPR78, showed suggestive association with lower odds of surviving to 144 the 90th percentile age in the discovery phase ($P = 7.0 \times 10^{-7}$) and was significant ($P = 0.01$) in the replication phase consisting of 373 cases and 2,271 controls. In the combined analysis of the discovery and replication phase the locus on chromosome 4 reached genome-wide significance (P

= 3.4×10^{-8}).

Gene-level association analysis identified genetically predicted expression of CRHR1 as significantly associated with survival to the 90th percentile age. Survival to the 90th and 99th percentile was genetically correlated with father's age at death, coronary artery disease and diabetes-related phenotypes providing insight in the complex genetic architecture of longevity.

The paper is well written, the statistical analyses are appropriate. It replicated well-known and produced new results of genetic analyses of human longevity. The results this paper will be of interest to others in the community and the wider field.

The authors of the paper represent an international group of highly qualified researchers studying genetics of human longevity. The paper written by specialists with such level of expertise would benefit from the discussion of conceptual framework for studying genetics of human longevity that could explain slow progress in the field, as well as the lack of replication of many findings detected in studies of independent populations. It would also strengthen the paper if the authors describe their view on promising directions of further research in this area. Several comments below deal with the issues that need clarifications.

Comments:

1. Line 206: Case and control definitions. It is unclear why the authors define control group using 60% percentile which corresponds to age 75 years in the 1920 birth cohort. Using this definition, the persons from the control groups in some studies included in this paper are likely to belong to younger birth cohorts which have better survival than participants of the 1920 birth cohort. This means that their 60% percentile will correspond to ages higher than 75 years. Our experience showed that the results of genetic analyses are sensitive to the age threshold used for specifying the control group. This sensitivity may partly be due to different histories of exposure to external environmental and living conditions, migrations patterns etc. experienced by different birth cohorts. The authors may miss some interesting findings by restricting themselves by the control group with 60% percentile. The discussion of this issue or the analysis of sensitivity of the research findings to the conditions defining the control group (if possible) would be beneficial for this paper.

2. Line 486: ...allele specific response to cellular stress were observed. Please reference paper where such connection has been described.

3. The reference to recent results of longevity analysis performed by LLFS group (Yashin et al., 2018. Journal of Gerontology Biological Sciences) would be relevant.

4. In the studies included in this paper the Quality Control (QC) procedures use p -value $\geq 10^{-4}$ - 10^{-6} for HW. One should not expect HW-equilibrium for longevity related SNPs in the "case" groups due to mortality selection. Using such p -values in QC may remove most interesting SNPs from the analysis. The discussion of this issue would be important in the paper.

Anatoliy Yashin

Reviewer #3:

Remarks to the Author:

The manuscript by Deelen et al. reported their meta-analysis study of longevity GWAS by applying a new definition of human longevity, i.e. surviving at or beyond the age corresponding to the 90th/99th survival percentile. The concept is clearly novel with general impact in the field, and the article is well-written and intelligibly discussed. I have a few major issues which I hope the authors can address.

1. By consensus percentile means the fraction of sample in the unit of percentage, and thus above 99th percentile in the context of this manuscript, for example, should indicate 1% of the cohort with longest life span. This is however not the case in the manuscript according to the numbers shown in all instances. In the abstract, the 90th and 99th percentile are 10,889 and 3,183, and for controls, supposedly below 60th percentile, the number is $N=23,212$. These are clearly not matching up if 1 percentile is equivalent to 1% of cohort. In some cohorts, such as CEPH, the N for

99th percentile is even larger than controls, which to me is very difficult to understand. Hence I would suggest authors to further clarify how the percentile groups are defined, and if possible provide histograms for age distributions for every cohort.

2. The novel identification of CRHR1 as a longevity-associated gene is, in my opinion, not solid and could be considered over-stated for several reasons: 1) the approach of using GTEx to convert GWAS association to gene expression association is only a prediction that has never been experimentally validated with RNA-seq data, not even in the original article. 2) The validity to apply such approach on age phenotype can be potentially problematic because age itself is a variable in the GTEx dataset. Although GTEx included age as an independent factor in their linear equation, such adjustment could be too simplistic as the impact of age on eQTL in the dataset has not been well characterized. 3) The founder of MetaXcan also provided PrediXcan, an alternative algorithm which is supposed to replicate MetaXcan result in ideal situation. The authors should report their result from this tool and confirm their finding.

3. The authors essentially reproduced known longevity loci from previous studies such as APOE, FOXO3, and CKDN2A/B. This is great but I believe the authors can make better efforts to take the full advantage of this large sample meta-analysis, for example to explore the impact of population background on the effect size of identified loci, or to determine whether the effect of these loci are linear throughout the entire aging process, or rather on a particular stage of aging.

Minor issues:

1. The usage of eMAC in EasyQC does not seem justified. The actual minor allele count in all cohorts are clearly available, and I do not see the point to make further adjustments which may distort the data. Using harmonic mean in the calculation of Effective N is always going to generate a smaller N, and consequentially, the actual allele count threshold is greater than reported, which leads to a better than actual QC result.

2. I strongly recommend authors to check the http links provided in the article. Many of them, such as indication of data availability, only link to the front page of the database and there is no indication how to find the specific dataset used in this manuscript. For another example, the link to UK National Archive directs to a non-existing page.

3. In discovery phase, there is no mention of an association analysis for 99th percentile vs dead control. Since 90th percentile vs dead control revealed new loci with suggestive $P < 1 * 10^{-7}$, 99th percentile vs dead control may reveal new loci. It will be informative to include it in the supplementary.

4. It is reported that 9 variants with suggestive P value were taken for replication, 3 of them are discovered in 90th percentile vs all controls/dead controls and six of them in 99th percentile vs all controls. Since 99th percentile phenotype is more extreme than that of 90th percentile, it is expected to detect variants discovered in GPR78, EFCC1 and CSMD1 in 99th percentile vs all controls. What are the results for these variants in 99th percentile vs controls and for 6 variants discovered in 99th vs all control in 90th vs control. I understand the sample size is smaller for 99th percentile but are the results even trend in the same direction.

5. Line #392 table reference is wrong. It should be "table 4" instead "table 5"

6. Table 1 and table S1: There is no consensus between the numbers in the two tables. Total N for replication phase and trans-ethnic phase should be included in Table 1. Summary statistics for the meta-analysis cohort is not mentioned in Table S1 (age-range, mean age, % of female). Numbers for 90th percentile and 99th percentile are swapped in Table 1 for AGES, CEPH, CHS,DKLS, DKLSII,GEHA-french, GEHA-Danish 100-plus/LASA/ADC, LLFS, LLS + GEHA Dutch, Longevity, MrOS, SOF cohorts. Total N needs to be changed accordingly

7. Table 4: legend does not specify the multiple testing p-value cut off and what highlighted p values indicate?

8. Controls were defined as individuals that died at or before the age at the 60th percentile or whose age at the last follow up visit was at or before the 60th percentile age. There is not enough explanation as to why 60th percentile or below is used to define controls and not 50th or 70th or between 50th and 60th percentile? Looking at the mean age and age-range of control cohorts, 3 cohorts have mean age below 30 with age range (0-65, 18-44, 16-42 years). The control definition seems to be too vague and includes any individual from newborn to the individual with age at 60th percentile. This may introduce bias. Moreover, at least some portion of these people may become centenarians which is a limitation of the study.

9. Based on Table S1 it looks like birth year cohorts for most of the cases and controls are different which may lead to confounding effects. It needs to be discussed as a limitation of the study.

10. For trans-ethnic meta-analysis- are the CLHLS sample definitions for cases and controls used here same as the original study (Zeng et al.,) It appears so based on the Table S1. Although IL-6 association is replicated and mainly driven by association in Asian population, it is not mentioned why the association for rs2440012 ANKRD20A9P is not replicated considering similar effect sizes for these two variants in Asian population.

11. The study does not report any sex-stratified analysis or even discuss about it. It would be informative to perform sex-stratified analysis to see if any gender dependent effects are observed.

Reviewers' comments:

Reviewer #1 (Remarks to the Author):

In a clean and tidy meta-analysis of longevity, looking at 90th and 99th percentiles of survivorship within each cohort, the authors confirm at unprecedented levels of statistical significance, the known deleterious effect of APOE e4 and identify a clear signal for an independent beneficial variant APOE e2 at this locus.

The authors go on to identify an association at GPR78 with lower levels of statistical significance and using gene level analyses to identify associations at CRHR1. They go onto show genetic correlations between longevity and a number of health related traits.

The combination of data using consistent phenotyping is a positive step forward. The statistical methods applied are well tried and appear appropriate in their application.

The identification of GPR78 is interesting. Whilst the authors are commendably transparent in the results used to make the inference, the combined evidence is of medium, rather than great strength. A simpler combined data presentation might make the paper clearer, without losing any robust important results. Robustness could be increased with a much bigger sample, but this is unrealistic.

Comments

Major

line 296: Nature Communications has an admirable policy on data disclosure, it would be helpful to review (a small sample of) the specific fields and field labels of the data proposed to be disclosed. Discovery and replication data should both be provided.

We were indeed planning to make our summary data available after publication, so the provided links were not yet functional. However, to accommodate the request of the Reviewer, we have decided to update the links so the data are now available through <https://www.longevitygenomics.org/downloads>.

~line 324

The separation into discovery and replication, with a relaxed threshold for discovery is not without precedent, but carries risk of Type 1 error. The authors argue that as only 3 tests of replication were performed for the 90th percentile, the 1% p-value for GPR78 passes Bonferoni adjustment. However, 11 replication tests were performed in total, and even if q-values were used (taking APOE out of consideration and removing duplicates), the full set of replication tests may mean the 5% (q or adjusted p) threshold is not exceeded. The relaxed discovery threshold and the less than conclusive replication suggest the GPR78 finding is interesting, but cannot yet be considered fully robust.

Furthermore, rough meta-analysis of discovery and replication suggest that OR=0.65 (0.56-0.75). Whereas the UKBB results the authors present suggest OR=0.96(0.84-1.12). This suggests to me the UKBB results contradict, rather than are merely underpowered. (Doubling effects UKBB for the effect of parent imputation, on the assumption this was not done already. Not doubling would have meant an even stronger contradiction.)

The authors should either (a) be more equivocal about the overall picture for GPR78; or (b) combine discovery and replication and then use UKBB as confirmation (failing at GPR78) (UKBB does appear powered to do the job for a reasonably large set of discovered SNPs, based on the above (UKBB SEs appear similar to discovery); or (preferably, but more work) meta-analyse (b) with UK Biobank too.

We agree with the Reviewer that it would be better to combine the genome-wide replication dataset with the discovery phase datasets (option b) and have done this accordingly. The reason that we did not do this before is that the replication dataset was added after we had already finished the discovery phase analyses (we were not aware of the existence of the replication dataset before). However, given that the phenotype used in the UK Biobank is different from the one we are using, we do not feel comfortable adding this cohort to the discovery phase as well. Instead we used this study to validate our findings. We have revised the manuscript based on the updated results.

We also agree that the evidence for the association of the *GPR78* locus with longevity is of medium strength due to the lack of evidence for association in the UK Biobank. Hence, we have mentioned in our Discussion section that this locus would benefit from replication in independent cohorts in the future (p. 22).

~line 376

Units in ST5 appear confused. For Timmers et al, the reported betas appear multiplied by 10. The reported betas were ln HR, so exp(reported beta) would seem the most comparable thing to report. For Pilling et al, the number reported is the effect size for rank normalised martingale residuals. This is not an HR and not comparable and should be reported as an effect size (eg beta) with legend explaining the scale used.

We would like to thank the Reviewer for spotting these errors. We have changed Supplementary Table 4 accordingly. The beta values reported by Timmers et al. were indeed multiplied by 10,¹ so for the calculation of the HR's we used "ln(beta/10)".

~line 479.

Some of the sense here is reasonable, but the most likely reason is that (as implicitly acknowledged), the authors do not have power to detect the presumable modest longevity effects of these loci, given the multiple testing and the power of their study. Consider the following line of reasoning using line 41 and 43 of table S5. LPA/CDKN2B appear an unambiguous lifespan locus. They also appear to be a longevity loci (OR=1.13/1.06 p=0.012/0.002), but allowing for multiple testing, the authors are not powered enough to confirm this.

Some discussion that the authors are not able to refute that these could well be real lifespan, and perhaps real (modest) longevity loci, (OR<1.1), that did not replicate due to lack of power and perhaps greater effect on mortality than longevity should be included.

Some discussion as to whether present lifespan studies appear to be more sensitive than present longevity studies in picking up longevity loci should be included (perhaps with the caveat that merely avoiding early death isn't the most biologically interesting form of longevity).

We agree with the Reviewer that the failure of replication (after adjustment for multiple testing) of some of the SNPs previously associated with parental lifespan may be due to their small effect size and the modest power of our study and that they may thus still be relevant for longevity. This was further supported by our observation that most of the loci previously associated with parental lifespan also showed a nominal significant association in the 90th percentile versus all controls dataset from the UK Biobank, indicating that the lack of replication may not be due to the use of a different phenotype. Hence, we have added a new Table (Supplementary Table 5) and made some textual changes to both the Results and Discussion sections to address this point (pp. 18-19 and pp. 23-24).

The use of consistent phenotyping is to be commended, but some discussion that there is more than one form of consistency (eg 85+, 95+), and that these might be biologically relevant, would be helpful

Although we agree with the Reviewer that there are also other forms of consistency that could be used to define a longevity phenotype, we feel that these are more prone to heterogeneity due to the differences in survival probabilities between birth cohorts. We added some text to the Discussion section (p. 27) in which we further clarify this issue.

Minor

~line 217. It would be nice to see a table or graph of the 90th and 99th percentile ages for all cohorts (survival, curves would be even better)

We provide the mean age and age range of the 90th and 99th percentile cases for each cohort in Supplementary Table S1.

~line 252 replace "heterogeneity" with "inflation"

This has been changed accordingly.

Line 327,330 : there is a confusion between variant/allele and polymorphism here. Perhaps just specify the allele at line 327?

This has been changed accordingly.

Reviewer #2 (Remarks to the Author):

The authors performed two meta-analyses of GWA studies of longevity applying a rigorous definition of cases and controls based on country-, sex-, and birth cohort-specific life tables. The discovery phase included 10,889/3,183 cases surviving at or beyond the age corresponding to the 90th/99th survival percentile, respectively, and 23,212 controls whose age at death or at last contact was at or below the age corresponding to the 60th survival percentile. Consistent with previous reports, rs429358 (ApoE ϵ 4) was associated with lower odds of surviving to the 90th and 99th percentile age.

The authors also detected a genome-wide significant association of rs7412 (ApoE ϵ 2) with higher odds of surviving to the 90th and 99th percentile age. One SNP, rs7676745, located on chromosome 4 near GPR78, showed suggestive association with lower odds of surviving to 144 the 90th percentile age in the discovery phase ($P = 7.0 \times 10^{-7}$) and was significant ($P = 0.01$) in the replication phase consisting of 373 cases and 2,271 controls. In the combined analysis of the discovery and replication phase the locus on chromosome 4 reached genome-wide significance ($P = 3.4 \times 10^{-8}$).

Gene-level association analysis identified genetically predicted expression of CRHR1 as significantly associated with survival to the 90th percentile age. Survival to the 90th and 99th percentile was genetically correlated with father's age at death, coronary artery disease and diabetes-related phenotypes providing insight in the complex genetic architecture of longevity.

The paper is well written, the statistical analyses are appropriate. It replicated well-known and produced new results of genetic analyses of human longevity. The results this paper will be of interest to others in the community and the wider field.

The authors of the paper represent an international group of highly qualified researchers studying genetics of human longevity. The paper written by specialists with such level of expertise would benefit from the discussion of conceptual framework for studying genetics of human longevity that could explain slow progress in the field, as well as the lack of replication of many findings detected in studies of independent populations. It would also strengthen the paper if the authors describe their view on promising directions of further research in this area.

During the revision of our manuscript we have included text that addresses some of these points. However, given that we substantially extended our Discussion and Results sections to address all comments of the Reviewers, we have not enough space left to elaborate on the direction of future research on the genetics of human longevity.

Several comments below deal with the issues that need clarifications.

Comments:

1.Line 206: Case and control definitions. It is unclear why the authors define control group using 60% percentile which corresponds to age 75 years in the 1920 birth cohort. Using this definition, the persons from the control groups in some studies included in this paper are likely to belong to younger birth cohorts which have better survival than participants of the 1920 birth cohort. This means that their 60% percentile will correspond to ages higher than 75 years. Our experience showed that the results of genetic analyses are sensitive to the age threshold used for specifying the control group. This sensitivity may partly be due to different histories of exposure to external environmental and living conditions, migrations patterns etc. experienced by different

birth cohorts. The authors may miss some interesting findings by restricting themselves by the control group with 60% percentile. The discussion of this issue or the analysis of sensitivity of the research findings to the conditions defining the control group (if possible) would be beneficial for this paper.

We acknowledge that we did not give enough explanation why we used the 60th percentile age as cut-off for controls and have therefore added some text to the Discussion section (p. 26). The reason is that we wanted to include as many controls as possible (preferably from the same study population), while leaving a large enough age gap between our cases and controls. We are aware that the majority of controls are coming from later birth cohorts than the cases and that this may lead to differences in survival. However, since we used birth cohort-specific percentiles we hope we have sufficiently taken this issue into account. We are also aware that some of the controls could still reach an age above the 90th/99th percentile and have acknowledged this as a limitation in the Discussion section (p. 27). On the other hand, given that misclassification of controls would lead to a reduction in the difference between cases and controls, this would actually strengthen our findings.

2.Line 486: ...allele specific response to cellular stress were observed. Please reference paper where such connection has been described.

The papers that have described this connection are placed after the next sentence (i.e. “Consistently, the longevity-associated alleles of all three SNPs were shown to induce *FOXO3* expression”). We did not put the references in twice, since the two sentences are connected to each other.

3. The reference to recent results of longevity analysis performed by LLFS group (Yashin et al., 2018. Journal of Gerontology Biological Sciences) would be relevant.

We have added this reference to the manuscript (p. 7). In addition, we have performed a look-up of rs1927465, which was shown to be genome-wide significantly associated with age at death in this paper, and have added the results to Supplementary Table S4. Sadly, this locus does not show any significance in our datasets. Moreover, we could not confirm the association of rs1927465 in the LLFS datasets that were used for this study ($P = 0.08$ (90th percentile) and $P = 0.07$ (99th percentile)).

4. In the studies included in this paper the Quality Control (QC) procedures use p-value $\geq 10^{-4}$ - 10^{-6} for HW. One should not expect HW-equilibrium for longevity related SNPs in the “case” groups due to mortality selection. Using such p-values in QC may remove most interesting SNPs from the analysis. The discussion of this issue would be important in the paper.

We may indeed have removed some genetic variants during the QC that were out of HWE in the cases due to the use of a more stringent HWE P-value cut-off (i.e. 1×10^{-4} - 1×10^{-6} versus, for example, 1×10^{-10}). However, most studies calculated the HWE P-value based on their whole dataset (i.e. cases + controls), so if these variants were only out of HWE in the cases they may still have been included in the analysis. We have made some textual changes to the Methods section (p. 10) to address the point of the Reviewer.

Anatoliy Yashin

Reviewer #3 (Remarks to the Author):

The manuscript by Deelen et al. reported their meta-analysis study of longevity GWAS by applying a new definition of human longevity, i.e. surviving at or beyond the age corresponding to the 90th/99th survival percentile. The concept is clearly novel with general impact in the field, and the article is well-written and intelligibly discussed. I have a few major issues which I hope the authors can address.

1. By consensus percentile means the fraction of sample in the unit of percentage, and thus above 99th percentile in the context of this manuscript, for example, should indicate 1% of the cohort with longest life span. This is however not the case in the manuscript according to the numbers shown in all instances. In the abstract, the 90th and 99th percentile are 10,889 and 3,183, and for controls, supposedly below 60th percentile, the number is N=23,212. These are clearly not matching up if 1 percentile is equivalent to 1% of cohort. In some cohorts, such as CEPH, the N for 99th percentile is even larger than controls, which to me is very difficult to understand. Hence I would suggest authors to further clarify how the percentile groups are defined, and if possible provide histograms for age distributions for every cohort.

It seems that the Reviewer has misinterpreted the way we defined our cases and controls. As mentioned in our Methods section we used cohort life tables from census data from the appropriate country, sex, and birth cohort. Hence, the number of selected cases and controls is defined by the ages of their birth cohort corresponding to the 60th or 90th/99th percentile age and is independent of the study population used (i.e. the number of controls and cases within a study population is not based on the percentiles of that specific population, but instead of their birth cohorts). As part of their recruitment protocol, many of the studies enrolled participants that were already relatively old at the time of recruitment. The Cardiovascular Health Study, for example, only included individuals that were above 65 years of age at the time of recruitment. The majority of these individuals subsequently survived past the 60th percentile age threshold of their respective birth cohorts (e.g. the maximum age was 84 for females from the 1925 birth cohort), resulting in a small number of controls (n = 558) in comparison to the number of cases (n = 905 (90th percentile)) for this study. We have made some textual changes to the Methods section (p. 9) to further clarify how we defined our cases and controls.

2. The novel identification of CRHR1 as a longevity-associated gene is, in my opinion, not solid and could be considered over-stated for several reasons: 1) the approach of using GTEx to convert GWAS association to gene expression association is only a prediction that has never been experimentally validated with RNA-seq data, not even in the original article. 2) The validity to apply such approach on age phenotype can be potentially problematic because age itself is a variable in the GTEx dataset. Although GTEx included age as an independent factor in their linear equation, such adjustment could be too simplistic as the impact of age on eQTL in the dataset has not been well characterized. 3) The founder of MetaXcan also provided PrediXcan, an alternative algorithm which is supposed to replicate MetaXcan result in ideal situation. The authors should report their result from this tool and confirm their finding.

We thank the Reviewer for the chance to discuss these important issues related to the genetic prediction of gene expression.

It is indeed correct that MetaXcan is a prediction-based approach and that the findings from this program will need experimental validation. Hence, we have mentioned that more functional work is required to determine the exact relationship between the altered genetically-driven tissue-specific expression of the identified genes and longevity in humans (p. 24). In the original PrediXcan paper from 2015,² the authors compared their elastic net models with the heritability of gene expression (i.e. the upper limit of the prediction performance) in the same cohort (DGN), based on whole-blood RNA sequencing data. This analysis showed that the average prediction R^2 for elastic net was close (0.137) to the average heritability (0.153). Additionally, the elastic net predictive performance reached or exceeded the lower bound of the heritability estimate for 94% of the genes. This indicates that the elastic net model used by PrediXcan is able to capture most of the heritable component of gene expression.

We agree with the Reviewer that it seems that the underlying GTEx models of MetaXcan might not have been adequately adjusted for age (i.e. they do not take non-linear age effects into account), which could be problematic for an age-related phenotype like longevity. Hence, we have added this as a limitation to our study in our Discussion section (pp. 24-25). However, MetaXcan has successfully been used to identify gene-level associations with age-related diseases and traits, such as Alzheimer's disease and age-related macular degeneration.³ We would also want to point out that we do not claim to be able to infer at what moment during lifespan genetically-driven gene expression might be associated with longevity.

The Reviewer suggested that we should try to replicate our MetaXcan findings using PrediXcan. Both methods use the exact same elastic net models to predict tissue-specific gene expression with genetic variants. However, an important difference between the two methods is that PrediXcan uses individual-level genotype data, while MetaXcan uses summary-level meta-analyses results. Hence, running PrediXcan, as the Reviewer suggested, would require obtaining individual-level genotype data from all contributing studies, which is not feasible due to data access restrictions on sharing of this data. This is precisely why the PrediXcan authors created MetaXcan. Moreover, in their original paper, the authors showed (using data from the Wellcome Trust Case Control Consortium) that results from PrediXcan and MetaXcan for bipolar disorder ($r^2 = 0.996$) and type 1 diabetes ($r^2 = 0.995$) are highly correlated. Therefore, we feel that the MetaXcan results are presentable without PrediXcan confirmation, similar to what has been done by others (see for example^{4,5}).

3. The authors essentially reproduced known longevity loci from previous studies such as APOE, FOXO3, and CKDN2A/B. This is great but I believe the authors can make better efforts to take the full advantage of this large sample meta-analysis, for example to explore the impact of population background on the effect size of identified loci, or to determine whether the effect of these loci are linear throughout the entire aging process, or rather on a particular stage of aging.

We agree with the Reviewer that it would be interesting to know if the replicated loci are affected by the genetic background of the population and if their effect becomes more prominent with increasing age. We therefore created forest plots (Supplementary Figure 5) for the two most interesting variants at these loci (rs2802292 and rs1556516) based on the 90th percentile cases versus all controls dataset. As depicted in this Figure, the effects of both variants fluctuate between the different cohorts and there seems to be no correlation with the genetic background of the included populations. However, for both loci, the odds of surviving to the 99th percentile age is higher than the odds of surviving to the 90th percentile age, indicating they likely affect both early and late-life mortality. We have made some textual

changes to the Results section to address this point (p. 18). We have to note that most of the individuals included in our analyses were from populations of European descent and there is limited availability of other data on the background of these individuals, which makes it hard to stratify the data within a given cohort based on population background. Moreover, the datasets used in this manuscript do not allow testing of effects throughout the entire aging process, given that we used a case-control design.

Minor issues:

1. The usage of eMAC in EasyQC does not seem justified. The actual minor allele count in all cohorts are clearly available, and I do not see the point to make further adjustments which may distort the data. Using harmonic mean in the calculation of Effective N is always going to generate a smaller N, and consequentially, the actual allele count threshold is greater than reported, which leads to a better than actual QC result.

The reason we used the eMAC including the Effective N instead of the Total N is that we wanted to balance the number of cases and controls in each study, as recommended in the paper describing EasyQC.⁶ We are aware that this leads to a more stringent filtering of variants, but we wanted to decrease the chance of false positive findings due to an imbalance between cases and controls. We have added some text to the Methods section (p. 11) in which we described the rationale behind the use of the eMAC. In addition, we realized that we forgot to mention that we used a more stringent eMAC filter (<25 instead of <10) for two of the datasets due to their large case:control imbalance (>1:20) as compared to the other datasets (all <1:10). We have now added this accordingly (see p. 11).

2. I strongly recommend authors to check the http links provided in the article. Many of them, such as indication of data availability, only link to the front page of the database and there is no indication how to find the specific dataset used in this manuscript. For another example, the link to UK National Archive directs to a non-existing page.

We would like to thank the Reviewer for spotting this. Since we were planning to make our summary data available after publication, we knew that the provided link was not yet functional. However, this should now be fixed. The link to the UK National Archive is correct, but due to a bug it is not correctly converted to the PDF. We will make sure to check it again once the manuscript is ready for typesetting.

3. In discovery phase, there is no mention of an association analysis for 99th percentile vs dead control. Since 90th percentile vs dead control revealed new loci with suggestive $P < 1 \times 10^{-7}$, 99th percentile vs dead control may reveal new loci. It will be informative to include it in the supplementary.

The inclusion of the 90th percentile versus dead control analysis was merely performed to look at the effect of the control definition on our results. Given that (1) the results of the two meta-analyses with different control groups were very similar and (2) the number of samples that could contribute to the 99th percentile versus dead control analysis is limited (resulting in low power to identify novel loci), we decided not to perform this analysis. For clarity, we decided to move the results from the 90th percentile cases versus dead controls analysis to the Supplementary Information.

4. It is reported that 9 variants with suggestive P value were taken for replication, 3 of them are discovered in 90th percentile vs all controls/dead controls and six of them in 99th percentile vs all controls. Since 99th percentile phenotype is more extreme than

that of 90th percentile, it is expected to detect variants discovered in GPR78, EFCC1 and CSMD1 in 99th percentile vs all controls. What are the results for these variants in 99th percentile vs controls and for 6 variants discovered in 99th vs all control in 90th control. I understand the sample size is smaller for 99th percentile but are the results even trend in the same direction.

We agree with the Reviewer that it would be good to show the complete results for all identified variants in both the 90th and 99th percentile analysis. We have therefore created a Figure (Supplementary Figure 2), in which the results of these variants are plotted for both analyses. As expected, most variants show stronger effects in the 99th percentile as compared to the 90th percentile analysis, indicating that the use of a more extreme phenotype results in stronger effects. We have also added some text to the Results section (p. 16) describing these results.

5. Line #392 table reference is wrong. It should be “table 4” instead “table 5”

We would like to thank the Reviewer for spotting this error. It has been changed accordingly.

6. Table 1 and table S1: There is no consensus between the numbers in the two tables. Total N for replication phase and trans-ethnic phase should be included in Table 1. Summary statistics for the meta-analysis cohort is not mentioned in Table S1 (age-range, mean age, % of female). Numbers for 90th percentile and 99th percentile are swapped in Table 1 for AGES, CEPH, CHS,DKLS, DKLSII,GEHA-french, GEHA-Danish 100-plus/LASA/ADC, LLFS, LLS + GEHA Dutch, Longevity, MrOS, SOF cohorts. Total N needs to be changed accordingly

It seems the Reviewer has misread the data provided in Supplementary Table 1, which was likely due to the fact that the columns for the 90th and 99th percentile cases were switched in comparison to Table 1. To avoid further confusion we have now switched the columns in Supplementary Table 1, so it now better matches with Table 1. In addition, we have added the total number of samples included in the Replication phase and the Trans-ethnic phase to Table 1.

7. Table 4: legend does not specify the multiple testing p-value cut off and what highlighted p values indicate?

This has been changed accordingly.

8. Controls were defined as individuals that died at or before the age at the 60th percentile or whose age at the last follow up visit was at or before the 60th percentile age. There is not enough explanation as to why 60th percentile or below is used to define controls and not 50th or 70th or between 50th and 60th percentile? Looking at the mean age and age-range of control cohorts, 3 cohorts have mean age below 30 with age range (0-65, 18-44, 16-42 years). The control definition seems to be too vague and includes any individual from newborn to the individual with age at 60th percentile. This may introduce bias. Moreover, at least some portion of these people may become centenarians which is a limitation of the study.

See our response to comment 1 of Reviewer 2.

9. Based on Table S1 it looks like birth year cohorts for most of the cases and controls

are different which may lead to confounding effects. It needs to be discussed as a limitation of the study.

The observation of the Reviewer is indeed correct; for most of the studies the controls and cases come from different birth cohorts. The reason for this is that most of the individuals from the same birth cohorts as the cases had already died before they could be included in the studies (i.e. provided DNA). Hence, we were forced to use controls from later birth cohorts. However, by using birth cohort-specific survival percentiles to define cases and controls (as clarified in our response to the first major comment of the Reviewer) we should have adjusted for this accordingly and, hence, this should not be a major limitation of our study.

10. For trans-ethnic meta-analysis- are the CLHLS sample definitions for cases and controls used here same as the original study (Zeng et al.,) It appears so based on the Table S1. Although IL-6 association is replicated and mainly driven by association in Asian population, it is not mentioned why the association for rs2440012 ANKRD20A9P is not replicated considering similar effect sizes for these two variants in Asian population.

We indeed used the same dataset as previously described.⁷ The reason that we were not able to replicate rs2440012 is that this variant did not pass quality control in the large majority of the included cohorts from populations of European descent and was thus not analysed in the trans-ethnic meta-analyses. We have added some text to the Results section explaining this (p. 17).

11. The study does not report any sex-stratified analysis or even discuss about it. It would be informative to perform sex-stratified analysis to see if any gender dependent effects are observed.

We agree with the Reviewer that a sex-stratified analysis may lead to identification of novel longevity loci that are currently missed. However, given (1) the limited number of identified suggestive significant loci for both the 90th and 99th percentile analysis, (2) our modest sample size, and (3) the lack of identified genome-wide significant sex-specific longevity loci, we decided not to perform sex-stratified analyses in this manuscript. We added some text to the Discussion section (p. 28) in which we acknowledged this as a limitation of our study.

References

1. Timmers PR, *et al.* Genomics of 1 million parent lifespans implicates novel pathways and common diseases and distinguishes survival chances. *Elife* **8**, (2019).
2. Gamazon ER, *et al.* A gene-based association method for mapping traits using reference transcriptome data. *Nat Genet* **47**, 1091-1098 (2015).
3. Barbeira AN, *et al.* Exploring the phenotypic consequences of tissue specific gene expression variation inferred from GWAS summary statistics. *Nat Commun* **9**, 1825 (2018).
4. Peng S, *et al.* Genetic regulation of the placental transcriptome underlies birth weight and risk of childhood obesity. *PLoS Genet* **14**, e1007799 (2018).
5. Lam M, *et al.* Large-Scale Cognitive GWAS Meta-Analysis Reveals Tissue-Specific Neural Expression and Potential Nootropic Drug Targets. *Cell Rep* **21**, 2597-2613 (2017).
6. Winkler TW, *et al.* Quality control and conduct of genome-wide association meta-analyses. *Nat Protoc* **9**, 1192-1212 (2014).
7. Zeng Y, *et al.* Novel loci and pathways significantly associated with longevity. *Sci Rep* **6**, 21243 (2016).

Reviewers' Comments:

Reviewer #1:

Remarks to the Author:

I congratulate the authors on their comprehensive response and amendments. In my opinion the manuscript is now suitable for publication.

Reviewer #2:

Remarks to the Author:

I am satisfied with the response of the authors

Reviewer #3:

Remarks to the Author:

This revision looks good. If anything I was about to question more on the choice of 60th percentile, but other reviewers have asked.

Reviewer #1 (Remarks to the Author):

I congratulate the authors on their comprehensive response and amendments. In my opinion the manuscript is now suitable for publication.

Reviewer #2 (Remarks to the Author):

I am satisfied with the response of the authors

Reviewer #3 (Remarks to the Author):

This revision looks good. If anything I was about to question more on the choice of 60th percentile, but other reviewers have asked.

We would like to thank all three Reviewers for their time and effort. Their comments and suggestions have been very constructive and have helped to improve our manuscript.

Editor

We therefore invite you to revise your paper one last time to include the discussion of future directions in longevity genetics research originally requested by Reviewer #2. We feel that this would be an important addition to the current work.

We have added an extra paragraph to the Discussion section (p. 22) in which we discuss future directions in longevity genetics research.